# JUST-IN-TIME PIECEWISE-LINEAR SEMANTICS FOR RELU-TYPE NETWORKS

## ABSTRACT

We present a JIT PL semantics for ReLU-type networks that compiles models into a guarded CPWL transducer with shared guards. The system adds hyperplanes only when operands are affine on the current cell, maintains global lower/upper envelopes, and uses a budgeted branch-and-bound. We obtain anytime soundness, exactness on fully refined cells, monotone progress, guard-linear complexity (avoiding global $\binom{k}{2}$), dominance pruning, and decidability under finite refinement. The shared carrier supports region extraction, decision complexes, Jacobians, exact/certified Lipschitz, LP/SOCP robustness, and maximal causal influence. A minimal prototype returns certificates or counterexamples with cost proportional to visited subdomains.

## 1 INTRODUCTION AND POSITIONING

**Problem & motivation.** For neural networks whose computation DAG is composed of affine modules (FC/conv/mean-pool/inference-time BN/residual) and pointwise gates (ReLU/Leaky-ReLU/PReLU/Abs/Max), each output coordinate is a *continuous piecewise-linear* (CPWL) function of the input. This makes a wide range of geometric and formal tasks—e.g., region extraction, gradients/Jacobians, decision boundaries, Lipschitz analysis, robustness/ equivariance/ equivalence checks, and intervention/repair—*exactly* expressible at the level of CPWL fragments. The bottleneck is *expression explosion*: if we expand the network as a symbolic max/+ tree layer by layer, the number of linear fragments and comparators can grow as $2^{\Theta(N)}$ in the number of gates, making the "compile first, solve later" pipeline infeasible even for medium-size models.

**Method overview.** We propose a unified symbolic carrier for CPWL networks: (i) a **Symbolic Weighted Transducer (SWT)** over the *function semiring* $\left(\mathrm{CPWL}^{\pm\infty}, \oplus = \max, \otimes = +\right)$, where edges carry *CPWL weights* and *polyhedral guards*; (ii) a **JIT-SWT** execution semantics that performs *on-demand refinement* instead of a static global expansion. JIT-SWT maintains a shared DAG of lazy expressions (an e-graph of `Max/Sum/Scale/Compose`) and a *global guard library*. It inserts threshold/comparator hyperplanes only when a visited GuardSet requires them, constructs local minimal common refinements and winner regions on-the-fly, and always tracks sound *anytime envelopes* $(\underline{A}, \overline{A})$ so that $\underline{A} \leq F \leq \overline{A}$ at all times. When all relevant faces inside a queried subdomain have been made explicit and the expression collapses to a single affine map, the envelopes meet and equal the true function—yielding *exactness on demand*.

**Contributions.**

1. **CPWL layer calculus and static SWT semantics.** We establish CPWL closure (AF-1..5) and an *equivalent* network→SWT compilation (SWT-1).

2. **JIT-SWT objects, invariants, and refiners.** We define the guard library/Guard-Sets and lazy e-graph; propose three refiners (ENSURE_SIGN, ENSURE_WINNER, ENSURE_COMMON_REFINE); and prove *DYN-1..6*: sound envelopes; exactness under local full refinement; no regression; budgeted linear upper bounds in #splits/#new guards; correctness of dominance pruning; and decidability under finite refinement.

3. **Decidable geometry & formal analysis on JIT.** We extract regions/gradients/Jacobians and compute extrema and *exact* Lipschitz constants; robustness/equivalence/equivariance follow via

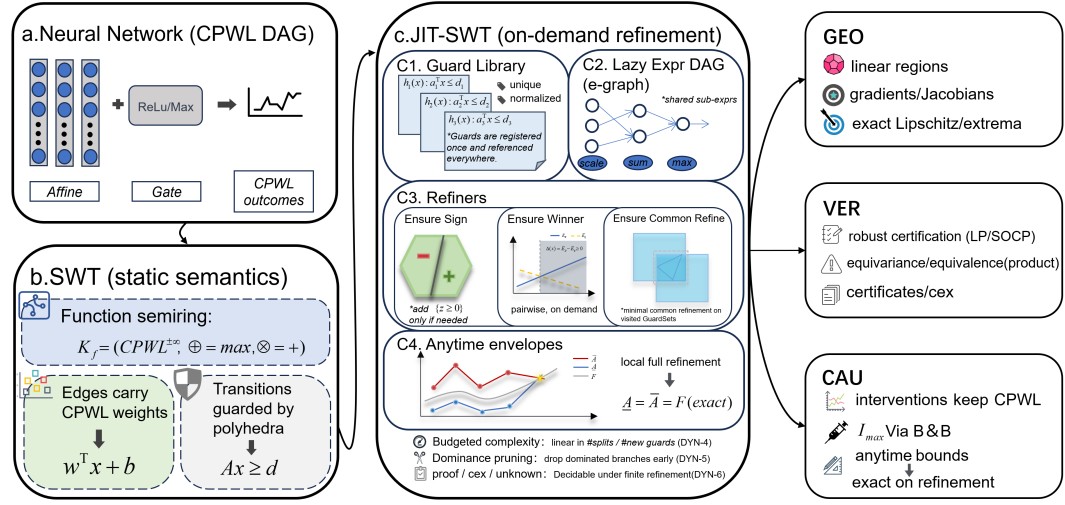

Figure 1: **Contribution map and framework overview.** Networks $\rightarrow$ SWT (static semantics) $\rightarrow$ JIT-SWT (on-demand refinement) $\rightarrow$ unified geometry/verification/causality with anytime certificates and exactness under local full refinement.

| Label | One-line meaning (full proofs in the appendix) |
| --- | --- |
| AF-1..5 | CPWL closure; homomorphism; continuity/convexity; a.e. differentiability; compositionality. |
| SWT-1 | Equivalent compilation: SWT equals the network pointwise. |
| SWT-3/4 | Decidable equivalence via common refinement; difference-region automata. |
| SWT-5 | Minimal guard complexity exists; decision is NP-hard. |
| DYN-1 | Sound anytime envelopes $\underline{A} \leq F \leq \overline{A}$. |
| DYN-2/3 | Exactness under local full refinement; progress never regresses. |
| DYN-4 | Budgeted linear upper bounds in #splits/#new guards. |
| DYN-5 | Correctness of dominance pruning. |
| DYN-6 | JIT-decidability under finite refinement; certs/counterexamples. |

Table 1: **Index of core statements.** Labels align with the theorem tags used throughout.

product constructions and branch-and-bound with certificates; causal interventions admit exact or anytime bounds.

4. **Small yet sufficient experiments.** FFN/CNN/GNN demonstrate machine-precision forward equivalence after local refinement and practical benefits: FFN—local Lipschitz correlates with FGSM success (Top-50 $100\%$ vs Bottom-50 $14\%$ at $\varepsilon=0.25$); CNN—$85\%$ translation-equivariance pass with failures at padding boundaries; GNN—permutation equivariance and Imax-guided ablations (Top-5 removal $-2.94\%$ accuracy; Bottom-5 $0\%$).

**Related work.** *Max-of-affine/tropical* methods target *convex* CPWL but do not directly capture general (possibly nonconvex) CPWL with shared guards (Butkovič, 2010). *Weighted automata* traditionally use discrete alphabets and scalar weights (Mohri, 2009); our SWT lifts weights to CPWL functions and augments edges with polyhedral guards. Neural verification via SMT/MILP/convex relaxations (Katz et al., 2017; Ehlers, 2017; Tjeng et al., 2019; Wong & Kolter, 2018; Bunel et al., 2020) is largely solver-centric and end-to-end, lacking a reusable, incrementally refiable *shared geometric object*. JIT-SWT serves as that carrier: properties reduce to finite affine comparisons/LP/SOCP on a *local* common refinement with anytime certificates, without global enumeration. Our on-demand refinement echoes *abstract interpretation*/*CEGAR* (Cousot & Cousot, 1977; Clarke et al., 2000), but lifted from program states to CPWL geometry with guard-driven predicates and certificate-producing solvers.

## 2 SETUP AND THE STATIC BASELINE: SWT TRUTH SEMANTICS

**CPWL and the function semiring (recap).** Let CPWL denote the class of continuous piecewise-linear real-valued functions over $\mathbb{R}^n$, and let $\mathrm{CPWL}^{\pm\infty}$ further include the everywhere $-\infty$ function. We work over the *function semiring*

$$K_f = \big(\mathrm{CPWL}^{\pm\infty}, \oplus, \otimes, \mathbf{0}, \mathbf{1}\big), \qquad (f \oplus g)(x) = \max\{f(x), g(x)\}, \quad (f \otimes g)(x) = f(x) + g(x),$$

with $\mathbf{0} \equiv -\infty$, $\mathbf{1} \equiv 0$. All network quantities (edge/node weights in the compilation graph) are CPWL functions, and all layerwise combinations are formed with pointwise $\max/+$.

**AF-1..AF-5 (statements only; proofs in App. B).**

- **AF-1 (well-definedness).** Under our component set (affine/conv/mean-pool/inference BN/residual; ReLU/LReLU/PReLU/Abs/Max) and DAG structure, every pre-activation is CPWL and every activation remains in $K_f$.

- **AF-2 (pointwise homomorphism).** Evaluating the expression at a point $x_0$ by the semiring homomorphism $h_{x_0} : f \mapsto f(x_0)$ reproduces the numerical forward pass.

- **AF-3 (continuity; sufficient convexity).** Outputs are CPWL and hence continuous; if all mixing coefficients entering each activation are nonnegative and the activation is convex and monotone (ReLU/LReLU/PReLU/Max), the network is convex. Abs preserves convexity only when applied directly to an affine atom.

- **AF-4 (a.e. differentiability).** CPWL functions are affine on a finite polyhedral complex and are differentiable Lebesgue-a.e.; Clarke subdifferentials at boundaries equal convex hulls of adjacent gradients.

- **AF-5 (closure under composition).** Finite compositions of "affine + the above gates" remain CPWL.

**Symbolic Weighted Transducer (SWT): static truth semantics.** We compile a network into a *guarded* symbolic weighted transducer

$$A = (Q, q_{\mathrm{in}}, F, E, \mathsf{G}, \mathsf{W}, K_f).$$

Here $Q$ are states (structural sites / template instances), $q_{\mathrm{in}} \in Q$ is the initial state, $F \subseteq Q$ are final states, and $E \subseteq Q \times Q$ are directed edges. Each edge $e \in E$ carries a *polyhedral guard* $\mathsf{G}(e) = \{x : Ax \le d\}$ in H-form drawn from a global guard library $\mathcal{H}$ (each normalized inequality is registered once), and a *CPWL weight* $\mathsf{W}(e) \in K_f$ (node weights similarly). A path $\pi = e_1 \dots e_T$ is feasible at $x$ if $x \in \bigcap_t \mathsf{G}(e_t)$, and its value is

$$\mathsf{val}_A(\pi, x) = \mathsf{W}(q_{\mathrm{in}})(x) \otimes \Big(\bigotimes_{t=1}^{T} \mathsf{W}(e_t)(x)\Big) \otimes \mathsf{W}(q_T)(x).$$

The SWT output is the semiring sum over feasible paths,

$$A(x) = \bigoplus_{\pi:\, q_{\mathrm{in}} \to F,\ x \models \pi} \mathsf{val}_A(\pi, x).$$

**Theorem 2.1** (SWT-1: equivalent compilation; proof in App. C.2)**.** *For any network $F$ built from our component set and any input $x$ in the domain, the SWT $A_F$ obtained by the rules below satisfies $A_F(x) = F(x)$.*

*One-line essence.* Affine branches compose by $\otimes(= +)$ along paths; gate choices and pooling reduce to $\oplus(= \max)$ over guarded winners, so path-wise accumulation and path-set maximization match the numerical forward semantics pointwise.

**Compilation rules at a glance (details in App. C.1).** We compile layer-by-layer while reusing the shared guard library $\mathcal{H}$ and never materializing global region partitions:

1. **Affine / residual / inference-time BN.** For a fragment $(C, w, b)$ (guard $C$, affine $x \mapsto w^\top x + b$), an affine map $y = Wx + b_0$ yields $(C, \tilde{w} = Ww, \tilde{b} = Wb + b_0)$. Inference-time BN is merged into the affine. Residual sums become $\otimes$-sums on the common guard $C$.

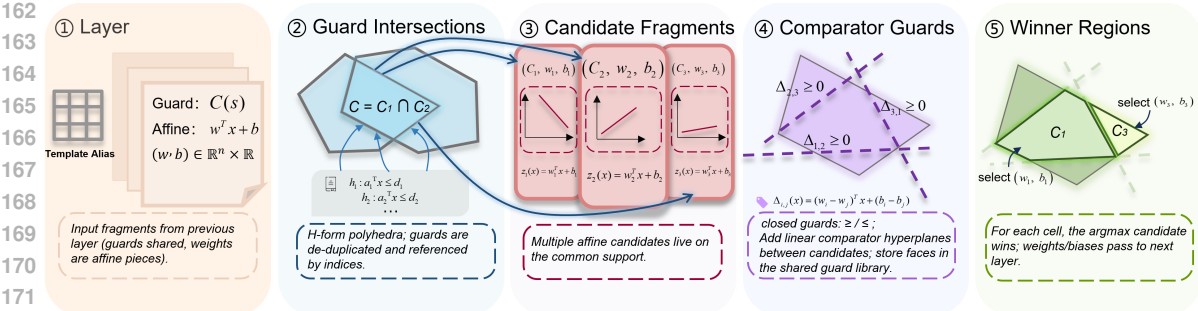

Figure 2: **Static SWT compilation pipeline.** Layer → guard intersections → candidate fragments → comparator guards → winner regions.

2. **ReLU / Leaky-ReLU / PReLU / Abs (pointwise gates).** Introduce two guarded branches on $z(x) = w^\top x + b$:

$$(\text{pos}) \ C^+ = C \cap \{z \geq 0\} \rightsquigarrow (w, b), \qquad (\text{neg}) \ C^- = C \cap \{z \leq 0\} \rightsquigarrow \begin{cases} (0, 0) & \text{ReLU} \\ (\alpha w, \alpha b) & \text{LReLU/PReLU} \\ (-w, -b) & \text{Abs} \end{cases}$$

using *closed* guards ($\geq, \leq$) to keep ties on both sides; $\oplus$ resolves equality.

3. **Pointwise Max / MaxPool.** Given candidates $\{(C_i, w_i, b_i)\}_{i=1}^k$ with common support $C_\cap = \bigcap_i C_i$, add comparators $\{(w_i - w_j)^\top x + (b_i - b_j) \geq 0\}$ to $\mathcal{H}$ and select the winner region

$$C_i^\star = C_\cap \cap \bigcap_{j \neq i} \{(w_i - w_j)^\top x + (b_i - b_j) \geq 0\}, \quad \text{carrying } (w_i, b_i).$$

4. **Convolution / mean-pool.** Instantiate a per-location *template* with *sliding-window guards* encoding stride/padding; share kernel parameters by aliasing to avoid numeric duplication. Mean-pool is affine.

5. **GNN (fixed graph).** Sum/mean aggregation compiles to sparse affine maps (template per node); max aggregation uses comparator guards; node-MLPs follow the same affine+gate rules.

6. **Normalization.** All guard rows are $\ell_2$-normalized; we adopt closed-set convention throughout.

**Size and acyclicity (informal SWT-2; constants in App. C.3).** Because the network DAG compiles forward without back-edges and guards are stored by *indices* into the shared library (we do not use regions as states), the SWT is acyclic; the number of states grows linearly with the number of affine sublayers and convolutional template instances, while the number of edges/guards is linear in these plus the number of *introduced comparator hyperplanes*. This avoids per-path duplication and isolates the exponential blow-up solely to the (optional) global enumeration of winner comparisons—which we *do not* perform in the static object.

## 3 DYNAMIC COMPILATION (JIT–SWT): OBJECTS, REFINERS, AND GUARANTEES

**Objects and invariants.** We endow the static SWT (§2) with a *JIT* (on-demand) execution semantics. Let $\mathcal{H} = \{h_\ell(x) : a_\ell^\top x \leq d_\ell\}_{\ell=1}^M$ be the global *guard library* where each normalized halfspace is registered once. A *GuardSet* is an ordered finite index set $S \subseteq \{1, \ldots, M\}$ representing the polyhedron $C(S) = \bigcap_{\ell \in S} \{x : a_\ell^\top x \leq d_\ell\}$. Lazy expressions (*scalar* case) are e-graph nodes with syntax

$$\texttt{Expr} ::= \texttt{Affine}(w, b) \mid \texttt{Sum}(\mathcal{E}) \mid \texttt{Max}(\mathcal{E}) \mid \texttt{Scale}(c, E) \mid \texttt{Bias}(b, E),$$

shared by structure hashing/union-find. Vector outputs use a componentwise family of scalar `Expr`s.

**Invariants (I–III).** *I: unique guards.* Each inequality in $\mathcal{H}$ appears once; edges store *indices* only (no explicit path intersections). *II: on-demand refinement.* New threshold/comparator hyperplanes

---

**Algorithm 1 JIT–SWT Branch-and-Bound (B&B) driver (anytime)**

---

1: **Input:** SWT $A$, domain $D$ as $S_0$, objective $g$ (e.g., spec margin or difference), budgets $B_{\text{split}}, B_{\text{guard}}$
2: $Q \leftarrow \{S_0\};$     $\text{CERT} \leftarrow \varnothing$         ▷ queue of active GuardSets; certificate store
3: **while** $Q \neq \varnothing$ and budgets not exhausted **do**
4:      $S \leftarrow \arg\max_{T \in Q} \text{UB}(g, T) - \text{LB}(g, T)$         ▷ max-gap policy
5:      **if** $\text{LB}(g, S) \geq 0$ **then** $\text{CERT} \leftarrow \text{CERT} \cup \{S\}$; $Q \leftarrow Q \setminus \{S\}$; **continue**
6:      **if** $\text{UB}(g, S) < 0$ **then return** COUNTEREXAMPLE from $\min_{x \in C(S)} g(x)$
7:      try ENSURE_WINNER / ENSURE_SIGN on $S$; else use ENSURE_COMMON_REFINE; push children to $Q$
8: **if** $Q = \varnothing$ **then return** PROOF with CERT
9: **else return** UNKNOWN with $S^\star = \arg\max_T \text{UB} - \text{LB}$

---

are inserted *only* on the *visited* GuardSet $S$, splitting $S$ locally into $S \cup \{\ell\}$ and $S \cup \{\bar{\ell}\}$; no global pre-partitioning. *III: anytime envelopes.* At all times we maintain

$$\underline{A}, \overline{A}: \quad \forall x \in \mathcal{D}, \qquad \underline{A}(x) \leq A(x) \leq \overline{A}(x),$$

where $A$ is the static SWT semantics (truth semantics). Envelopes are propagated compositionally from local *LB/UB* on Exprs (below).

**LB/UB oracle (structure rules).** For any GuardSet $S$ and expression $E$ we compute interval $[\text{LB}(E, S), \text{UB}(E, S)]$ sound for $C(S)$: [1]

$$\text{LB}(\texttt{Affine}(w, b), S) = \min_{x \in C(S)} w^\top x + b, \quad \text{UB}(\texttt{Affine}(w, b), S) = \max_{x \in C(S)} w^\top x + b \text{ (LP);}$$

$$\text{LB}(\texttt{Sum}(\{E_k\}), S) = \sum_k \text{LB}(E_k, S), \quad \text{UB}(\texttt{Sum}(\{E_k\}), S) = \sum_k \text{UB}(E_k, S);$$

$$\text{LB}(\texttt{Scale}(c, E), S) = \begin{cases} c\,\text{LB}(E, S), & c \geq 0 \\ c\,\text{UB}(E, S), & c < 0 \end{cases}, \quad \text{UB}(\texttt{Scale}(c, E), S) = \begin{cases} c\,\text{UB}(E, S), & c \geq 0 \\ c\,\text{LB}(E, S), & c < 0 \end{cases};$$

$$\text{LB}(\texttt{Bias}(b, E), S) = \text{LB}(E, S) + b, \quad \text{UB}(\texttt{Bias}(b, E), S) = \text{UB}(E, S) + b;$$

$$\text{LB}(\texttt{Max}(\{E_k\}), S) = \max_k \text{LB}(E_k, S), \quad \text{UB}(\texttt{Max}(\{E_k\}), S) = \max_k \text{UB}(E_k, S).$$

**Atomic refiners (3–5 lines each; full details in App. D.2).** *(i)* ENSURE_SIGN($z$=Expr, $S$)*:* if $\text{UB}(z, S) \leq 0$ commit the negative branch; if $\text{LB}(z, S) \geq 0$ commit the positive; otherwise insert the threshold guard $\{z \geq 0\}$ into $\mathcal{H}$ (if absent) and split $S$ into $S^{\pm}$.
*(ii)* ENSURE_WINNER($\{E_i\}, S$)*:* compute $[\text{LB/UB}(E_i, S)]$; prune dominated candidates $\{i : \max_j \text{LB}(E_j, S) \geq \text{UB}(E_i, S)\}$; if a unique $i^\star$ remains with $\text{LB}(E_{i^\star}, S) \geq \max_{j \neq i^\star} \text{UB}(E_j, S)$, commit $i^\star$; else add one comparator $\{E_p \geq E_q\}$ chosen by a gap heuristic and split $S$.
*(iii)* ENSURE_COMMON_REFINE($S; S_1, S_2$)*:* incrementally insert faces from $(S_1 \cup S_2) \setminus S$ until the relevant sub-expressions are comparable/addable on each child.

**Anytime envelopes and monotonicity.**

**Lemma 3.1** (Monotone envelope updates; App. D.1)**.** *Adding guards (shrinking $C(S)$) or tightening local bounds makes $\underline{A}$ nondecreasing and $\overline{A}$ nonincreasing pointwise; in particular $\underline{A} \leq A \leq \overline{A}$ is preserved.*

**Theorem 3.2** (DYN-1: sound anytime envelopes; App. D.1)**.** *At any time during JIT execution, $\underline{A}(x) \leq A(x) \leq \overline{A}(x)$ holds over $\mathcal{D}$.*

**Anytime exactness and no regression.**

**Definition 3.3** (Local full refinement)**.** A GuardSet $S$ is *locally fully refined* if (i) all gate/comparator faces relevant to $S$ have been added to $\mathcal{H}$, and (ii) each output Expr collapses to a *single affine* on $C(S)$.

---

[1] Upgrades to exact LP/SOCP bounds are permitted anytime; proofs of soundness/monotonicity are in App. D

**Theorem 3.4** (DYN-2: exactness on $C(S)$; App. D.2). *If $S$ is locally fully refined, then on $C(S)$ we have $\underline{A} = \overline{A} = A$.*

**Theorem 3.5** (DYN-3: progress never regresses; App. D.2). *Refining a GuardSet $S$ either keeps it as a tighter leaf (with tighter LB/UB) or splits it into children; previously exact regions remain exact and are never invalidated.*

**Budgeted complexity upper bounds.**

**Theorem 3.6** (DYN-4; App. D.4). *Let $B_{\textit{split}}$ be the number of local splits and $B_{\textit{guard}}$ the number of newly inserted hyperplanes (beyond the initial library size $|\mathcal{H}_0|$). Then: (i) the number of active GuardSets is $\leq 1 + B_{\textit{split}}$; (ii) $|\mathcal{H}| \leq |\mathcal{H}_0| + B_{\textit{guard}}$; (iii) the SWT node/edge count grows $O(N_{\mathrm{lin}} + T_{\mathrm{conv}} + B_{\textit{guard}})$; (iv) the number of LP/SOCP calls is $O(B_{\textit{split}} + B_{\textit{guard}} + |Q|)$ under per-step constant-candidate policies.*

**Dominance pruning.**

**Proposition 3.7** (DYN-5; App. D.3). *On $S$, if $\max_{x \in C(S)}(E_\pi(x) - E_{\pi'}(x)) \leq 0$ (e.g., obtained by an LP), then candidate $\pi$ can never win any max/gate on $C(S)$ and may be safely pruned from $S$'s candidate set.*

**Decidability and tri-valued answers.**

**Theorem 3.8** (DYN-6: JIT-decidability; App. D.3). *For properties expressible with finitely many affine and second-order cone constraints, if the JIT strategy is* fair *(every feasible, not-yet fully refined GuardSet is eventually selected) and only finitely many faces can be added, then the B&B process decides the property: it returns either a* proof *(certificate $\mathrm{LB} \geq 0$ on a finite cover), or a* counterexample *(witness from $\mathrm{UB} < 0$ on some $S$), otherwise* unknown *under budget—always sound thanks to Thm. D.4.*

**Static–dynamic pointwise equivalence.**

**Definition 3.9** (Complete refinement cover). A finite family $\mathcal{S} = \{S_t\}$ is a complete refinement cover of $D$ if $D \subseteq \bigcup_t C(S_t)$ and each $S_t$ is locally fully refined.

**Theorem 3.10** (DYN-7; App. D.3). *If $D$ admits a complete refinement cover $\mathcal{S}$, then $\forall x \in D$: $A_{\textit{JIT}}(x) = A_{\textit{stat}}(x) = F(x)$.*

**Notes on ties and numerics.** All threshold/comparator guards are treated as closed ($\geq, \leq$), which guarantees coverage of $z{=}0$ events; implementations may use a small tolerance $\tau$ to handle floating-point ties without affecting the formal semantics (App. G).

## 4 DECIDABLE ANALYSIS AND FUNCTION GEOMETRY ON JIT

We now show how geometry (regions/gradients/Jacobians, extrema/Lipschitz) and formal tasks (verification, equivariance, causality) become *decidable* on top of the JIT–SWT semantics from §3. Throughout, proofs and full algorithms are deferred to Appendix E–F.

### 4.1 REGIONS, GRADIENTS, AND JACOBIANS (GEO-1/4)

**Active fragments and local exactness.** Recall (Def. 3.3) that a GuardSet $S$ is *locally fully refined* when all faces relevant to $S$ are present and every output expression collapses to a single affine map on $C(S)$. In this case we write the scalar output as $F(x) = w_S^\top x + b_S$ on $C(S)$ (or componentwise $F_i(x) = w_{S,i}^\top x + b_{S,i}$).

**Theorem 4.1** (GEO-1: on-demand region extraction; App. E.1). *Running JIT refinement restricted to a domain $D$ returns a set of* active fragments *$\{(S_\rho, w_\rho, b_\rho)\}_\rho$ whose (relative-interior) union covers the visited portion of $D$, and on each $C(S_\rho)$ the function equals the affine law $F(x) = w_\rho^\top x + b_\rho$. If refinement continues until $D$ is covered by finitely many locally fully refined GuardSets, this yields a complete CPWL region table for $F$ on $D$.*

**Gradients/Jacobians and boundary representatives.** On the relative interior of each $C(S_\rho)$, the gradient/Jacobian is constant: $\nabla F(x) = w_\rho$ (or $J_F(x) = J_\rho$ for vector outputs). At polyhedral boundaries, $F$ is not differentiable in general; we use Clarke subdifferentials.

**Theorem 4.2** (GEO-4: a.e. exact Jacobian; App. E.1). *Lebesgue-a.e.* $x \in D$ *lies in the relative interior of some refined* $C(S_\rho)$, *and the JIT extractor returns* $\nabla F(x)$ *(or* $J_F(x)$*) exactly. If* $x$ *lies on a boundary, a selection rule (e.g., minimum-norm element from the convex hull of adjacent gradients, obtained by a tiny QP) yields a representative in the Clarke subdifferential.*

**Practical readout on JIT.** Once $S$ is locally fully refined, we read $(w_S, b_S)$ by summing affine atoms along the unique acyclic path fragments that remain active on $C(S)$. For vector outputs we assemble $J_S$ row-wise.

## 4.2 Extrema and Lipschitz (GEO-2/5)

**Anytime maxima/minima.** Let $D$ be a convex domain (box/polytope/$\ell_p$ ball). JIT–B&B maintains

$$\underline{M} = \max_{S \subseteq D} \mathrm{LB}(F, S), \qquad \overline{M} = \max_{S \subseteq D} \mathrm{UB}(F, S),$$

which satisfy $\underline{M} \leq \max_{x \in D} F(x) \leq \overline{M}$ and tighten monotonically (Thm. D.4). When all visited $S$ are locally fully refined, each subproblem reduces to affine maximization on a convex set, solved exactly by LP or SOCP.

**Theorem 4.3** (GEO-2: exact extrema after local refinement; App. E.3). *If* $D$ *is covered by locally fully refined GuardSets* $\{S\}$, *then*

$$\max_{x \in D} F(x) = \max_S \max_{x \in C(S) \cap D} \left( w_S^\top x + b_S \right),$$

*with LP for polytope/$\ell_\infty$/$\ell_1$, SOCP for $\ell_2$ or polytope$\cap \ell_2$; the minima are analogous.*

**Lipschitz constants.** For scalar $F$, the exact $L_p$ on $D$ equals the maximum dual-norm of local gradients. For vector $F : \ell_p \to \ell_r$, it equals the maximum operator norm of local Jacobians.

**Theorem 4.4** (GEO-5: piecewise maximum formula; App. E.4). *If* $D$ *is covered by locally fully refined* $\{S\}$, *then*

$$L_p(F; D) = \max_{S : C(S) \cap D \neq \varnothing} \|w_S\|_{p^*}, \qquad L_{p \to r}(F; D) = \max_{S : C(S) \cap D \neq \varnothing} \|J_S\|_{p \to r}.$$

*Before full refinement, JIT provides anytime bounds* $\underline{L} \leq L \leq \overline{L}$ *by combining exact values on refined leaves with sound relaxations on unresolved leaves (LP/SOCP based).*

## 4.3 Verification, Equivariance, and Causality (VER/CAU)

**Property language and product construction.** We consider properties expressed by finite combinations of: input constraints $x \in D$ (box/polytope/$\ell_p$ ball), output thresholds/intervals $F_k(x) \leq u$, $F_k(x) \geq \ell$, classification margins $F_y - \max_{j \neq y} F_j \geq \gamma$, and relational constraints $|F(x) - F'(x)| \leq \varepsilon$. Each atomic constraint is compiled into a *counterexample transducer* and composed (product) with the SWT of $F$, yielding a CPWL objective $g$ whose violation corresponds to counterexamples. JIT–B&B on $g$ returns a tri-valued answer with certificates (Thm. D.11).

**Theorem 4.5** (VER-1: decidable specification checking; App. F.1). *For the above property language, JIT–B&B returns* TRUE *(with per-leaf LB certificates),* FALSE *(with a witness point from LP/SOCP on some leaf), or* UNKNOWN *under budget—always sound and complete under finite refinement.*

**Robustness certification.** For a labeled sample $(x_0, y)$, define the margin $g(x) = F_y(x) - \max_{j \neq y} F_j(x)$. Over $B_p(x_0, \epsilon) \cap D$, JIT refines until $g$ is affine on leaves and solves $\min g(x)$ by LP/SOCP per leaf. The minimum bound across leaves certifies robustness; a negative minimum yields a counterexample.

**Theorem 4.6** (VER-2: robustness via JIT; App. F.2). *On* $B_p(x_0, \epsilon) \cap D$, *the JIT procedure produces an anytime lower bound on the margin and, upon local full refinement, the* exact *minimum. Hence "$g \geq \gamma$" is decidable under finite refinement with a certificate or a witness.*

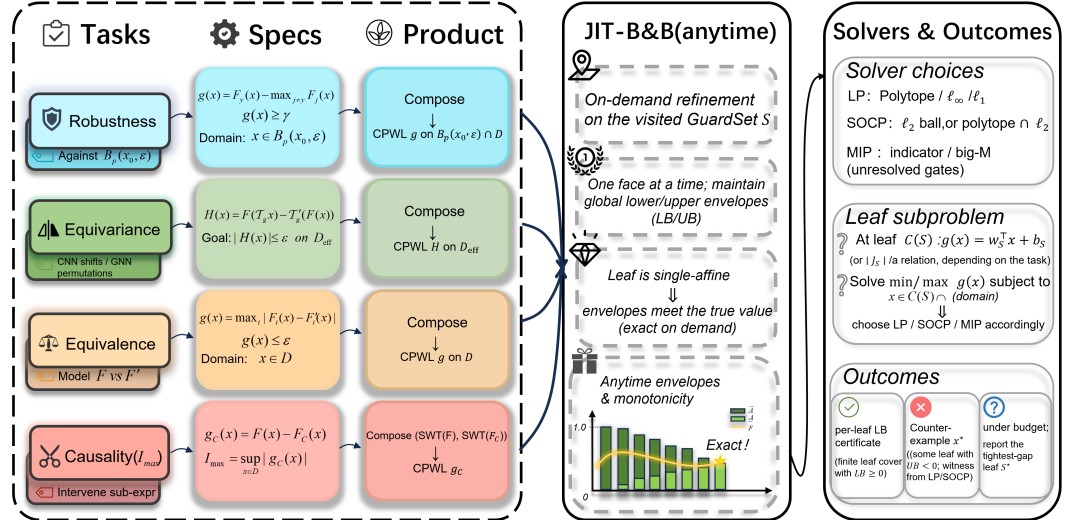

Figure 3: **Task–spec–solver map.** Properties are compiled into CPWL objectives via product constructions and discharged on JIT leaves by small LP/SOCPs (and MIP when indicator constraints are explicitly encoded). The JIT envelopes provide anytime certificates; local full refinement gives exact answers.

**Equivariance checks.** Given input transform $\mathcal{T}_g$ and output transform $\mathcal{T}_g'$ (e.g., CNN translations within an effective domain, or GNN permutations), we check $H(x) = F(\mathcal{T}_g x) - \mathcal{T}_g'(F(x))$. Running JIT–B&B on $|H|$ over the appropriate domain yields either a uniform upper bound $\leq \varepsilon$ with certificates, or an explicit violating input.

**Theorem 4.7** (VER-3: (approximate) equivariance; App. F.3)**.** *Over the effective domain, JIT decides exact equivariance ($H \equiv 0$) or $\varepsilon$-equivariance ($\|H\|_\infty \leq \varepsilon$) under finite refinement, with counterexamples or per-leaf bounds as certificates.*

**Causal interventions and maximal influence.** Consider an intervention that replaces a sub-expression in the DAG by a CPWL policy $P_c$, producing $F_C$ (App. F.4). Since CPWL is closed under the operations we use, $g_C(x) = F(x) - F_C(x)$ is CPWL. The *maximal causal influence* on $D$ is

$$I_{\max}(C; D) = \sup_{x \in D} |g_C(x)|,$$

which JIT computes by running B&B on $g_C$ and $-g_C$ with anytime bounds $\underline{I} \leq I_{\max} \leq \overline{I}$ and exact value upon full refinement.

**Theorem 4.8** (CAU-1/2: decidable interventions; App. F.4)**.** *For CPWL interventions $C$ and convex $D$, JIT–B&B yields sound anytime bounds on $I_{\max}(C; D)$ and the exact value once refined leaves cover $D$.*

**Notes on solvers and domains.** Affine maximization over boxes/polyhedra uses LP; over $\ell_2$ balls or polytope$\cap\ell_2$ uses SOCP; pure $\ell_2$ admits the closed form $w^\top x_0 + b + \epsilon \|w\|_2$. All calls are *local* to refined leaves; hence the global complexity follows the budgeted bounds of Thm. D.13.

## 5 EXPERIMENTS

We report three small yet sufficient studies that exercise JIT–SWT on FFNs, CNNs, and GNNs, focusing on (i) *forward equivalence* to the original PyTorch models after local refinement, and (ii) one geometry/formal task per setting. To save space we summarize the key numbers here; the representative visuals are in **Figs. 5, 7, 12** and full setup is deferred to the appendix H.

**FFN (MNIST): local Lipschitz & robustness.** For correctly classified test samples, we JIT-compile a local affine law $x \mapsto Wx + b$ per input and take $\|W\|_2$ as the *local* Lipschitz. We find

| Task | Forward equiv. | Geometry / statistic | Property outcome | Notes |
|------|----------------|----------------------|------------------|-------|
| FFN (MNIST) | avg. $1.13\times10^{-6}$ | local-$L$: Top/Bottom 6.60/2.23 | FGSM@$\varepsilon$=0.25: 100% vs 14% | $L_{\text{upper}}$=22.93 |
| CNN (CIFAR-10) | avg. $1.36\times10^{-7}$ | pass rate 0.85 over 8 shifts | 58 failures; concentrated at padding boundaries | Heatmap (Figure 7) |
| GNN (Karate) | avg. $2.17\times10^{-6}$ | perm.-equiv. mean err. $6.80\times10^{-15}$ | Remove Top-5 Imax $\downarrow$ 2.94%; Bottom-5 $\downarrow$ 0% | Bars (Figure 12) |

Table 2: **One-page summary.** Forward equivalence is at machine precision across all settings. Each geometry/formal task is discharged on JIT leaves by small LP/SOCPs, with anytime bounds and exactness upon local refinement.

*machine-precision* forward equivalence (avg. abs. error $1.13\times10^{-6}$) and a wide spread of local-$L$ (most between 2–7; see Figure 5). Sorting by local-$L$, the *Top-50* most sensitive points exhibit **100%** FGSM success at $\varepsilon$=0.25, vs. **14%** on the *Bottom-50*, confirming a strong correlation between local sensitivity and adversarial fragility.

**CNN (CIFAR-10 subset): translation equivariance.** On a small conv net with stride= 1, padding= 1, we verify forward equivalence on random samples (avg. abs. error $1.36\times10^{-7}$), then test 8 integer shifts $\Delta \in \{-1, 0, 1\}^2 \setminus \{(0,0)\}$. The *equivariance pass rate* is **85%**. Failures (58 total) concentrate at diagonal shifts (Figure 7), consistent with boundary effects from zero padding that propagate through ReLU and global pooling.[2]

**GNN (Karate Club): permutation equivariance & causal Imax.** For a 2-layer GCN we obtain near-exact forward equivalence (mean error $2.17\times10^{-6}$) and near-exact permutation equivariance over 50 random permutations (mean error $6.80\times10^{-15}$). We then compute *Imax* per hidden channel by intervening $h^{(2)}_{:,k} \leftarrow 0$ and running JIT–B&B over $\|\cdot\|_\infty$-bounded input neighborhoods. Channels ranked *Top-5* by Imax cause a **2.94%** accuracy drop when removed (vs. **0%** for *Bottom-5*); see Figure 11 for the distribution and Figure 12 for ablation bars.

**Takeaways.** (i) JIT–SWT attains precise *per-sample* linearization that is numerically identical to the original networks after local refinement; (ii) geometry is actionable—local Lipschitz pinpoints adversarially fragile points; (iii) JIT formal analysis scales to structure: translation equivariance violations are localized to padding faces; (iv) causal Imax separates crucial GNN channels from redundant ones and predicts ablation impact.

## 6   CONCLUSION AND LIMITATIONS

**Summary.** We introduced *SWT*, a symbolic carrier over the CPWL (max,+) semiring with polyhedral guards, and *JIT–SWT*, an on-demand semantics that shares an expression DAG, inserts faces only on visited cells, and maintains sound anytime envelopes. Geometry and formal tasks—region/gradient/Jacobian extraction, extrema and exact/certified Lipschitz, robustness/equivalence/equivariance, and causal interventions—reduce to small LP/SOCP subproblems on local common refinements, yielding certificates or counterexamples without global enumeration. Compact FFN/CNN/GNN case studies demonstrate machine-precision agreement and practical value.

**Limitations.** (i) Guarantees cover DAGs with affine modules and pointwise max-type gates; multiplicative modules (attention, LSTM/GRU) and training-time BN fall outside CPWL and need sound PL relaxations. (ii) Decidability assumes *finite* refinements and a *fair* selection policy; under budgets we return a sound tri-valued answer (TRUE/FALSE/UNKNOWN). (iii) Influence-type upper bounds derived from single-fragment linearization are non-certified unless the leaf is fully refined or tightened by a guard-aware refinement.

**Future work.** Certified PL lifts for multiplicative modules; compositional verified training/repair; parallel and learned refinement; matrix-free Jacobian extraction for large conv/GNNs; richer property languages (temporal/relational) and tighter anytime relaxations with certificates.

---

[2]Ablations with different padding fill values are in the appendix; the conclusion is robust.

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

# A    Notation, Domains, and the Guard Library

This appendix fixes notations, domain conventions, and the canonicalization rules for the global *guard library* used by SWT and JIT–SWT. All statements here are design-level and compatible with the truth semantics in §2 and the JIT semantics in §3.

**Basic sets and operators.**    For a set $S \subseteq \mathbb{R}^n$, let $\mathrm{ri}(S)$, $\partial S$, and $\mathrm{aff}(S)$ denote its relative interior, boundary, and affine hull. For a family $\{S_i\}$, write $\bigcap S_i$ and $\bigcup S_i$ for intersection and union. The indicator $\mathbb{I}[\cdot]$ takes values in $\{0, 1\}$.

## A.1    Domains and norms

**Definition A.1** (Input domains). We work with convex input domains $D \subseteq \mathbb{R}^n$ of the following types:

1. **Box/polytope (H–form).** $D = \{x \in \mathbb{R}^n : A_D x \leq d_D\}$ with $A_D \in \mathbb{R}^{m \times n}$.

2. **$\ell_p$ balls.** $D = B_p(x_0, \epsilon) = \{x : \|x - x_0\|_p \leq \epsilon\}$ for $p \in \{1, 2, \infty\}$.

When $D$ is not polyhedral (e.g., $p = 2$), we treat $D$ as a second–order cone constraint in local LP/SOCP subproblems; only *linear* faces are stored in the global guard library (§A.3).

**Definition A.2** (Norms and duals). For $p \in [1, \infty]$, let $\| \cdot \|_p$ be the vector $p$–norm and $p^*$ its dual with $1/p + 1/p^* = 1$ (by convention $1/\infty = 0$). For a matrix or linear map $J$, $\|J\|_{p \to r} = \sup_{\|v\|_p = 1} \|Jv\|_r$. We use the standard closed forms when applicable: $\|J\|_{1 \to 1} = \max_j \sum_i |J_{ij}|$, $\|J\|_{\infty \to \infty} = \max_i \sum_j |J_{ij}|$, and $\|J\|_{2 \to 2} = \sigma_{\max}(J)$.

## A.2    Polyhedral guards in H–form and closed–set semantics

A (closed) halfspace is encoded by a normalized pair $(a, d) \in \mathbb{R}^n \times \mathbb{R}$ as

$$h(x) \ : \ a^\top x \leq d.$$

A polyhedron in H–form is an intersection of finitely many closed halfspaces. We always interpret gate thresholds and comparator faces with *closed* inequalities ($\geq$ and $\leq$ on appropriate sides), so that "ties" (equality cases) belong to both sides and are resolved semantically by $\oplus = \max$ in the function semiring.

## A.3    The global guard library $\mathcal{H}$: canonicalization and uniqueness

**Normalization.**    Each linear inequality is stored once in a global library $\mathcal{H} = \{h_\ell(x) : a_\ell^\top x \leq d_\ell\}_{\ell=1}^M$ under the following canonicalization:

1. **Row normalization:** $\|a_\ell\|_2 = 1$.

2. **Sign canonicalization:** if the first nonzero entry of $a_\ell$ is negative, multiply both sides by $-1$ to make it nonnegative; this makes $(a, d) \sim (\alpha a, \alpha d)$ with $\alpha > 0$ share a unique representative.

3. **Tie policy:** for a threshold $\{z \geq 0\}$ we register *both* orientations $\{z \geq 0\}$ and $\{z \leq 0\}$ as distinct indices, so equality is covered on either side. We denote the reverse orientation of $\ell$ by $\bar{\ell}$.

**Comparator faces.**    For two affine forms $f(x) = w_f^\top x + b_f$ and $g(x) = w_g^\top x + b_g$, the comparator face $\{f \geq g\}$ is represented by $\big((w_f - w_g), (b_f - b_g)\big)$ and normalized as above before insertion.

**Storage discipline.**    Edges in SWT/JIT store *only indices* into $\mathcal{H}$; intersections are not materialized on edges (no "region as state"), cf. §2.3.

## A.4 GUARDSETS, FEASIBILITY, AND LEAF SEMANTICS

**Definition A.3** (GuardSet). A *GuardSet* is a finite ordered index set $S \subseteq \{1, \ldots, M\}$ representing the polyhedron

$$C(S) = \bigcap_{\ell \in S} \{x : a_\ell^\top x \leq d_\ell\}.$$

We write $S \cup \{\ell\}$ (resp. $S \cup \{\bar{\ell}\}$) for adding an oriented face.

**Definition A.4** (Feasibility cache). We maintain a cache feas : $S \mapsto \{\mathsf{unknown}, \mathsf{infeas}, \mathsf{feas}\}$, queried/updated by a single LP feasibility test for $C(S)$. Certificates (a feasible point or an infeasibility proof from the solver) may be stored alongside for reuse.

**Definition A.5** (Leaves and active cover). At any time, the JIT refinement maintains a finite set of *leaf* GuardSets $\mathcal{L} \subset 2^{\{1,\ldots,M\}}$ such that: (i) every visited $x$ belongs to at least one feasible $C(S)$ with $S \in \mathcal{L}$, and (ii) whenever a leaf $S$ is split along a (possibly new) face $\ell$, $S$ is *removed* from $\mathcal{L}$ and replaced by its children $S^+ = S \cup \{\ell\}$ and $S^- = S \cup \{\bar{\ell}\}$ that remain feasible.

*Remark* A.6 (Anytime envelopes are taken over *leaves*). All lower/upper bounds $\mathrm{LB}(E, S)$ and $\mathrm{UB}(E, S)$ used to assemble the anytime envelopes $(\underline{A}, \overline{A})$ in §3 are aggregated only over *current leaves* $S \in \mathcal{L}$ containing the query point $x$. Removing a parent from $\mathcal{L}$ upon splitting ensures the monotonicity $\underline{A} \uparrow$, $\overline{A} \downarrow$ proved in §3 (DYN–1/3).

## A.5 LAZY EXPRESSION DAG (E–GRAPH): SYNTAX, SEMANTICS, AND SHARING

**Syntax.** Scalar expressions are formed by the grammar

$$\mathtt{Expr} ::= \mathtt{Affine}(w, b) \mid \mathtt{Sum}(\mathcal{E}) \mid \mathtt{Max}(\mathcal{E}) \mid \mathtt{Scale}(c, E) \mid \mathtt{Bias}(b, E),$$

where $\mathcal{E}$ is a finite multiset of subexpressions, $c \in \mathbb{R}$, and $\mathtt{Affine}(w, b)(x) = w^\top x + b$. Vector outputs are tuples of scalar $\mathtt{Expr}$s.

**Pointwise semantics.** For $x \in \mathbb{R}^n$, the denotation $[\![E]\!](x)$ is defined recursively by

$$[\![\mathtt{Affine}(w, b)]\!](x) = w^\top x + b, \quad [\![\mathtt{Bias}(b, E)]\!](x) = [\![E]\!](x) + b,$$

$$[\![\mathtt{Scale}(c, E)]\!](x) = c \cdot [\![E]\!](x), \quad [\![\mathtt{Sum}(\{E_i\})]\!](x) = \sum_i [\![E_i]\!](x),$$

$$[\![\mathtt{Max}(\{E_i\})]\!](x) = \max_i [\![E_i]\!](x).$$

These agree with the function semiring $(\mathrm{CPWL}^{\pm\infty}, \oplus, \otimes)$ via $\oplus = \max$ and $\otimes = +$ (cf. §2).

**Sharing and congruence.** Expressions are stored in an *e–graph* with hash–consing and congruence closure:

1. **Affine interning.** Two affine atoms are identical iff their $(w, b)$ are componentwise equal (theory level) or equal within a fixed tolerance $\tau$ (implementation level; does not affect the closed–set semantics).

2. **AC canonicalization.** Children of $\mathtt{Sum}/\mathtt{Max}$ are stored in a deterministic order (e.g., lexicographic by node id) so that $\mathtt{Sum}(E_1, E_2)$ and $\mathtt{Sum}(E_2, E_1)$ share; likewise for $\mathtt{Max}$.

3. **Common subexpression elimination.** Any structurally equal subterm is stored once and referenced by pointers; this enables whole–graph updates when a subterm's bound is tightened.

**Comparator nodes and winners.** Comparator faces introduced by $\mathtt{Max}$ (and gate sign tests) are not embedded as Boolean nodes; instead they are materialized as linear faces registered in $\mathcal{H}$ (§A.3) when a visited leaf $S \in \mathcal{L}$ requires disambiguation. Winner selection is then expressed by restricting to the corresponding children $S \cup \{\ell\}$ or $S \cup \{\bar{\ell}\}$.

## A.6 TIES, FLOATING POINT TOLERANCE, AND DETERMINISM

All theoretical statements use closed guards and exact reals. Implementations may adopt a small tolerance $\tau$ (e.g., $10^{-7}$) to decide numeric equalities; when $|z(x)| \leq \tau$ at a hinge, both orientations are registered and explored under JIT, preserving soundness of the envelopes in §3.

## A.7 Summary table of symbols

| Symbol | Type | Meaning |
|---|---|---|
| $D$, $\mathcal{D}_{\text{in}}$ | set | Input domain / global domain (§A.1) |
| $B_p(x_0, \epsilon)$ | set | $\ell_p$ ball (SOCP when $p=2$) |
| $\|\cdot\|_p$, $p^*$ | number | Vector norm and its dual (§A.2) |
| $\|J\|_{p\to r}$ | number | Operator norm of a linear map $J$ |
| $\mathcal{H}$ | library | Global guard library (§A.3) |
| $h_\ell$ | halfspace | Guard $\{a_\ell^\top x \le d_\ell\}$; reverse index $\bar{\ell}$ |
| $S$ | set of ids | GuardSet; $C(S) = \bigcap_{\ell \in S} h_\ell$ (§A.3) |
| $\text{feas}(S)$ | enum | Feasibility cache (§A.4) |
| $\mathcal{L}$ | set | Current leaves (active cover; §A.5) |
| $\texttt{Expr}$ | DAG node | Lazy scalar expression (§A.5) |
| $\underline{A}, \overline{A}$ | function | Anytime envelopes (aggregated over leaves) |

# B  Proofs for AF-1..AF-5

This appendix supplies full proofs for the AF-series statements used in the main text: CPWL closure (AF–1), pointwise homomorphism (AF–2), continuity and sufficient convexity (AF–3), a.e. differentiability and Clarke subgradients (AF–4), and closure under composition (AF–5). We respect the closed-guard convention (both sides of hinges kept) fixed in §A.

## B.1  Preliminaries on CPWL functions and polyhedral complexes

**Definition B.1** (CPWL). *A function $f : \mathbb{R}^n \to \mathbb{R}$ is continuous piecewise linear (CPWL) if there exists a finite polyhedral complex $\mathcal{C} = \{R_\rho\}$ that covers $\mathbb{R}^n$ such that for each cell $R_\rho$, $f(x) = w_\rho^\top x + b_\rho$ holds on $R_\rho$ and adjacent cells agree on their shared faces (continuity).*

**Lemma B.2** (Finite overlays). *Let $\mathcal{C}_1, \ldots, \mathcal{C}_k$ be finite polyhedral complexes in $\mathbb{R}^n$. Then the common refinement $\mathcal{C} = \text{refine}(\mathcal{C}_1, \ldots, \mathcal{C}_k)$ obtained by intersecting all cells is a finite polyhedral complex.*

*Proof.* Each $\mathcal{C}_i$ is finite and closed under taking faces. Intersections of finitely many polyhedra are polyhedra, and the set of all nonempty intersections of finitely many cells is finite. Face-compatibility follows from face-compatibility of the inputs and distributivity of intersections over faces. $\square$

**Lemma B.3** (Basic CPWL closure). *If $f, g$ are CPWL on $\mathbb{R}^n$, then $f + g$ and $\max\{f, g\}$ are CPWL. If $A \in \mathbb{R}^{m \times n}, b \in \mathbb{R}^m$, then $x \mapsto f(Ax+b)$ is CPWL on $\mathbb{R}^m$.*

*Proof.* Write $f = w_\rho^\top x + b_\rho$ on cells $R_\rho$ of $\mathcal{C}_f$ and $g = u_\sigma^\top x + c_\sigma$ on cells $S_\sigma$ of $\mathcal{C}_g$. On the overlay $\mathcal{C} = \text{refine}(\mathcal{C}_f, \mathcal{C}_g)$, both are affine, hence $f+g$ is affine cell-wise. For $\max\{f, g\}$, further refine by the family of linear comparators $\{(w_\rho - u_\sigma)^\top x + (b_\rho - c_\sigma) \ge 0\}$; on each side of each comparator the winner is fixed and the result is affine. Finiteness is preserved by Lemma B.2. For composition with $Ax+b$, pull back each cell $R_\rho$ by $(Ax+b)$; preimages of polyhedra by an affine map are polyhedra, and continuity is preserved, so $f \circ (Ax+b)$ is CPWL on a finite complex. $\square$

## B.2  AF-1: Well-definedness (CPWL closure of the component set)

**Theorem B.4** (AF–1). *Under the component and DAG assumptions (affine/conv/mean-pool/inference BN/residual; ReLU/LReLU/PReLU/Abs/Max), every pre-activation is CPWL and every activation remains in $K_f = (\text{CPWL}^{\pm\infty}, \oplus = \max, \otimes = +)$.*

*Proof. Affine, residual, mean-pool, inference-time BN.* Each is an affine map in the input; convolution with fixed stride/padding is a linear map, hence affine. (Inference BN $y = \gamma \odot \frac{x-\mu}{\sigma} + \beta$ is affine with fixed $\mu, \sigma, \gamma, \beta$.) By Lemma B.3, affine maps preserve CPWL.

*Pointwise gates.* Let $z$ be a CPWL pre-activation.

$$\text{ReLU}(z) = \max\{0, z\}, \quad \text{LReLU}_\alpha(z) = \max\{z, \alpha z\} \ (\alpha \in [0, 1]), \quad \text{PReLU}_\alpha(z) = \max\{z, \alpha z\} \ (\alpha \geq 0),$$

$$\text{Abs}(z) = \max\{z, -z\}, \qquad \text{Max}(z_1, \ldots, z_k) = \max_i z_i.$$

Each right-hand side is a finite max of CPWL functions (affine multiples of $z$ or a finite family $\{z_i\}$), hence CPWL by Lemma B.3. The closed-guard convention ensures ties are covered on both sides and continuity is kept.

*DAG composition.* Layerwise application of the above operations along a DAG composes CPWL maps with affine maps and finite maxima; closure follows from Lemma B.3. $\square$

### B.3 AF-2: POINTWISE HOMOMORPHISM

**Theorem B.5** (AF–2)**.** *For any input $x_0$, evaluation $h_{x_0} : K_f \to (\mathbb{R}, \max, +)$ given by $h_{x_0}(f) = f(x_0)$ is a semiring homomorphism. Therefore the SWT expression obtained from a network evaluates to the same numeric forward value at $x_0$.*

*Proof.* By definition, $h_{x_0}(f \oplus g) = (f \oplus g)(x_0) = \max\{f(x_0), g(x_0)\} = \max\{h_{x_0}(f), h_{x_0}(g)\}$, and $h_{x_0}(f \otimes g) = (f \otimes g)(x_0) = f(x_0) + g(x_0) = h_{x_0}(f) + h_{x_0}(g)$. Also $h_{x_0}(\mathbf{0}) = -\infty$ and $h_{x_0}(\mathbf{1}) = 0$. Hence $h_{x_0}$ is a semiring homomorphism.

A network compiled to an SWT is a finite DAG of $K_f$-weighted nodes/edges combined by $\oplus$ (for selections/max) and $\otimes$ (for branch sums). Evaluating the SWT via $h_{x_0}$ pushes evaluation through the DAG and returns exactly the numeric forward pass (sum along the active branches; max across competing branches). Induction on the topological order of the DAG completes the proof. $\square$

### B.4 AF-3: CONTINUITY AND SUFFICIENT CONVEXITY

**Theorem B.6** (AF–3: continuity)**.** *Every network output coordinate is CPWL and hence continuous on $\mathbb{R}^n$.*

*Proof.* AF–1 shows all coordinates are CPWL. CPWL functions are continuous by definition of polyhedral complexes with face-compatible affine pieces. $\square$

**Theorem B.7** (AF–3: sufficient convexity)**.** *Suppose that at every layer, each activation $\phi$ is convex and nondecreasing (e.g., ReLU, LReLU with $\alpha \in [0, 1]$, PReLU with $\alpha \geq 0$, pointwise Max), and the mixing coefficients feeding each activation are entrywise nonnegative. Then each output coordinate is convex. If $\text{Abs}$ is used, convexity is preserved only when $\text{Abs}$ is applied directly to an affine pre-activation.*

*Proof.* Let $x \mapsto Ax + b$ be an affine pre-activation. Convexity is preserved under affine precomposition and nonnegative linear combinations: if $u_j$ are convex and $\lambda_j \geq 0$, then $\sum_j \lambda_j u_j$ is convex. If $\phi$ is convex and nondecreasing, then $\phi \circ u$ is convex whenever $u$ is convex. These are standard closure rules. Start from the input (convex), propagate through affine layers (convexity preserved), through nonnegative summations (residual/additive merges), and through convex nondecreasing activations (ReLU/LReLU/PReLU/Max), convexity is preserved at each step. For $\text{Abs}(t) = \max\{t, -t\}$: it is convex in $t$, but not monotone; in general $\text{Abs} \circ u$ need not be convex if $u$ is nonconvex CPWL. When $u$ is affine, $\text{Abs} \circ u$ is convex; this is the stated boundary case. $\square$

### B.5 AF-4: A.E. DIFFERENTIABILITY AND CLARKE SUBDIFFERENTIALS

**Theorem B.8** (AF–4)**.** *Let $f$ be CPWL on $\mathbb{R}^n$. Then $f$ is differentiable Lebesgue-a.e., and on each cell $R_\rho$ of a defining complex $\mathcal{C}$ we have $f(x) = w_\rho^\top x + b_\rho$ with $\nabla f(x) = w_\rho$ for all $x \in \text{ri}(R_\rho)$. At a boundary point $x \in \partial R_\rho$, the Clarke subdifferential is the convex hull of the neighboring gradients:*

$$\partial^C f(x) = \text{conv}\{ w_\rho : x \in R_\rho \}.$$

*Proof.* The union of the relative interiors of the finitely many cells covers $\mathbb{R}^n$ up to a set contained in a finite union of affine hyperplanes (cell boundaries), which has Lebesgue measure zero. On each relative interior $f$ is affine, hence differentiable with constant gradient $w_\rho$. The Clarke subdifferential of a locally Lipschitz function equals the closed convex hull of the set of limiting gradients; for CPWL the only limits arise from adjacent cells, yielding the stated formula. $\square$

### B.6 AF-5: CLOSURE UNDER FINITE COMPOSITIONS

**Theorem B.9** (AF–5)**.** *Any finite composition of layers built from affine maps, residual (additive) merges, and pointwise gates* $\{\mathrm{ReLU}, \mathrm{LReLU}, \mathrm{PReLU}, \mathrm{Abs}, \mathrm{Max}\}$ *yields a CPWL map.*

*Proof.* By AF–1 each individual layer maps CPWL inputs to CPWL outputs. By Lemma B.3, affine pre/post-composition, finite sums, and finite maxima preserve CPWL. Induction on the number of layers (topological order of the DAG) proves that the overall composition is CPWL. $\square$

**Remarks on convolution and pooling.** A discrete convolution with fixed kernel, stride, and padding is a linear operator on the vectorized input; mean-pooling is also linear. Therefore they are covered by the affine-case arguments above. Max-pooling is a finite pointwise maximum and falls under Lemma B.3. Inference-time batch normalization is affine.

**Closed guards and ties.** All thresholds/comparators are treated with closed inequalities ($\geq, \leq$) on both sides, so $z{=}0$ lies in both guarded branches. This preserves continuity and ensures that $\oplus = \max$ resolves ties without leaving gaps, consistent with the semantics used in the experiments and implementation.

## C SWT SEMANTICS AND EQUIVALENT COMPILATION (SWT-1/2/3/4/5)

This appendix formalizes the compilation of a CPWL network into a guarded *Symbolic Weighted Transducer* (SWT), proves pointwise equivalence (SWT–1), states size and acyclicity bounds with explicit counting conventions (SWT–2), gives common refinement / difference-region constructions and correctness (SWT–3/4), and provides an NP-hardness reduction for minimal-guard realizations (SWT–5).

### C.1 FORMAL COMPILATION RULES

**Object model and counting conventions.** We compile the network DAG to an acyclic SWT $A = (Q, q_{\mathrm{in}}, F, E, \mathsf{G}, \mathsf{W}, K_f)$ with:

- **States** ($Q$). One state per *structural site*: post-layer sites for affine modules (Dense/BN/MeanPool/Residual-add), post-gate sites (ReLU/LReLU/PReLU/Abs/Max), and per-template sites for convolution/GNN (defined below). We never use "region as state".

- **Edges** ($E$). A dataflow edge ($u \to v$) becomes one SWT edge unless $v$ is a gate, in which case guarded multi-edges are created (two for two-way gates; $k$ for Max-of-$k$).

- **Guards** ($\mathsf{G}$). Each edge stores an *index set* (GuardSet) into the global library $\mathcal{H}$; intersections are formed on-the-fly, not stored explicitly (cf. App. A).

- **Weights** ($\mathsf{W}$). CPWL weights in the function semiring $K_f = (\mathrm{CPWL}^{\pm\infty}, \oplus = \max, \otimes = +)$; along a path, weights compose by $\otimes(= +)$; across alternative paths they combine by $\oplus(= \max)$.

**Affine / residual / mean-pool / inference-time BN.** Let a fragment $(C, w, b)$ denote guard $C$ and affine map $x \mapsto w^\top x + b$. For $y = Wx + b_0$ we emit $(C, \tilde{w}, \tilde{b})$ with $\tilde{w} = Ww$, $\tilde{b} = Wb + b_0$. Residual $\mathsf{Sum}(x^{(1)}, x^{(2)})$ contributes one post-sum state and two incoming edges; weights add by $\otimes$ on the common guard. Inference-time BN is merged into affine.

**Two-way gates (ReLU/LReLU/PReLU/Abs).** Given pre-activation $z(x) = w^\top x + b$ on $C$, create two guarded edges to the post-gate site (closed guards on both sides):

$$\mathsf{pos:} \quad C^+ = C \cap \{z \geq 0\} \ \rightsquigarrow \ (w, b),$$

$$\mathsf{neg:} \quad C^- = C \cap \{z \leq 0\} \ \rightsquigarrow \ \begin{cases} (0,0) & \text{ReLU} \\ (\alpha w, \alpha b) & \text{LReLU/PReLU } (\alpha \geq 0) \\ (-w, -b) & \text{Abs} \end{cases}$$

Insert the (normalized) threshold face $\{z \geq 0\}$ into $\mathcal{H}$ (both orientations are registered; see App. A.3).

**Pointwise Max / MaxPool.** For candidates $\{(C_i, w_i, b_i)\}_{i=1}^k$ with $C_\cap = \bigcap_i C_i$, insert pairwise comparators $\{(w_i - w_j)^\top x + (b_i - b_j) \geq 0\}$ to $\mathcal{H}$ and create $k$ edges to the post-Max site, guarded by the winner regions

$$C_i^\star = C_\cap \ \cap \ \bigcap_{j \neq i} \{(w_i - w_j)^\top x + (b_i - b_j) \geq 0\}.$$

**Convolution (template aliasing; sliding-window guards).** For a conv layer with kernel $K \in \mathbb{R}^{C_{\text{out}} \times C_{\text{in}} \times h \times w}$, stride $(s_x, s_y)$, padding $(p_x, p_y)$, output spatial size $(H_{\text{out}}, W_{\text{out}})$, we instantiate one site per output channel-location $r = (c, i, j) \in \{1..C_{\text{out}}\} \times \{1..H_{\text{out}}\} \times \{1..W_{\text{out}}\}$, with *template aliasing* of weights: each site references the same $K$ by index. The receptive field constraint is captured by a rectangular sliding-window guard $C_r = \{x : \ M_r x = \text{vec}(x[\text{patch}(i, j)])\}$ which in H-form is the intersection of the input domain with the valid-range halfspaces implied by stride/padding (i.e., pixels outside the padded canvas are fixed to the padding value and realized as affine shifts). The linear map of conv at $r$ is an affine atom $(w_r, b_r)$ obtained by composing im2col with $K_{c,:,:,:}$; see App. I for a matrix-free variant.

**GNN on a fixed graph.** For an aggregation layer on $G = (V, E)$ with normalized adjacency $\hat{A}$ and output channels $C_{\text{out}}$, we instantiate $|V| \cdot C_{\text{out}}$ sites, each referencing shared parameters (template aliasing). Sum/mean aggregations compile to sparse affine maps $(\hat{A} \otimes I)$; Max aggregation uses comparator guards across neighbor terms. Per-node MLPs follow the affine+gate rules above.

*Remark* C.1 (No region as state; shared guard library). Edges store only guard indices. Winner regions and guard intersections are materialized on demand (JIT) or symbolically specified (static), but states are never duplicated per region. This is essential for the size bounds in §C.3.

## C.2 Proof of SWT-1 (pointwise equivalence)

**Theorem C.2** (SWT–1). *For any input $x$, the SWT $A_F$ compiled by §C.1 satisfies $A_F(x) = F(x)$.*

*Proof sketch with a worked miniature.* **Homomorphism.** By AF–2, evaluation $h_x : K_f \to (\mathbb{R}, \max, +)$ preserves $\oplus/\otimes$. Hence it suffices to check that each module compiles to an expression over $K_f$ whose $h_x$-evaluation equals the numeric forward of that module at $x$.

**Affine and residual.** Along a path, $\otimes$ accumulates affine weights by addition, matching numeric sums. A residual merge with fan-in two contributes two incoming edges whose $h_x$-values add (same $C$), equal to the module's numeric sum.

**Two-way gates and Max.** The closed guards ensure that at any $x$ exactly the feasible *winner* edge(s) remain; the outgoing value is the $\oplus$ over those edges, which equals the ordinary max (or gate rule) at $x$.

**Miniature for parallel addition.** Consider $y = f_1(x) + f_2(x)$ realized by two branches merging into a post-sum state $q$. Compilation creates edges $e_1 : q_{f_1} \to q$ and $e_2 : q_{f_2} \to q$ with weights $\mathsf{W}(e_i) = f_i$. There is a *single* path from $q_{\text{in}}$ to $q$ passing through the structure of $f_1$ and another passing through $f_2$ composed in series via an intermediate "sum" node where we use $\otimes$ to combine the two branch contributions on the *same* path (not via $\oplus$ across paths). Thus $\text{val}(\pi, x) = f_1(x) + f_2(x)$ by $\otimes$, while $\oplus$ across *alternative* guarded paths is reserved solely for max-type choices. This construction (explicit in the compiler IR) eliminates any ambiguity between sum and max.

**Convolution / GNN.** Each template instance is an affine map (linear with bias); template aliasing shares parameters but does not affect per-instance semantics. Therefore, $h_x$ on the compiled instance equals the numeric conv/GNN output at that site.

**Induction on the DAG.** Process nodes in topological order; at each step, the compiled fragment preserves the module's semantics at $x$. Hence $A_F(x) = F(x)$ globally. $\qquad\square$

*Remark* C.3 (Empirical alignment). On FFN/CNN/GNN models, the compiled SWT/JIT forward agrees with PyTorch forward to machine precision (mean absolute errors around $10^{-6} \sim 10^{-7}$), matching the predicted pointwise equivalence. See the per-sample error plots and tables in the experiment report (Task 1/2/3). *This is not part of the proof, but confirms the implementation adheres to the semantics.*

## C.3  SWT-2: ACYCLICITY AND SIZE BOUNDS WITH EXPLICIT CONSTANTS

**Counting parameters.**  Let

$$N_{\text{aff}} := \text{number of affine sites (Dense/BN/MeanPool/Residual-post)},$$

$$N_{\text{gate2}} := \text{number of scalar two-way gates (ReLU/LReLU/PReLU/Abs)},$$

$$\mathcal{V}_{\max} := \{v : \text{pointwise Max site of arity } k_v\}, \quad K_{\max} := \sum_{v \in \mathcal{V}_{\max}} k_v,$$

$$\Gamma_{\max} := \sum_{v \in \mathcal{V}_{\max}} \binom{k_v}{2}, \qquad \Gamma^{\pm}_{\max} := 2\,\Gamma_{\max} \ \text{(both orientations)},$$

$$T_{\text{conv}} := \sum_{\ell \in \text{conv layers}} C^{(\ell)}_{\text{out}} H^{(\ell)}_{\text{out}} W^{(\ell)}_{\text{out}},$$

$$T_{\text{gnn}} := \sum_{\ell \in \text{MP layers}} |V|\, C^{(\ell)}_{\text{out}}.$$

Assume each affine/conv/GNN site has in-degree 1 except residual-post sites whose in-degree is their fan-in (typically 2).

**Proposition C.4** (Acyclicity). *If the computation graph is a DAG, the compiled SWT is acyclic.*

*Proof.* Compilation preserves edge directions and introduces no back-edges; gate splits and winner guards do not add cycles, only parallel guarded edges. States correspond to post-module/template sites in the original topological order. $\qquad\square$

**Theorem C.5** (Size bounds with explicit constants). *Under the counting above, the following hold for the* static *SWT:*

$$\boxed{|Q| \ \leq \ 1 \ + \ N_{\text{aff}} \ + \ T_{\text{conv}} \ + \ T_{\text{gnn}} \ + \ |\mathcal{V}_{\max}| \ + \ N_{\text{gate2}}}$$

$$\boxed{|E| \ \leq \ N_{\text{aff}} \ + \ T_{\text{conv}} \ + \ T_{\text{gnn}} \ + \ (\textstyle\sum \text{residual fan-ins}) \ + \ 2N_{\text{gate2}} \ + \ K_{\max}}$$

$$\boxed{|\mathcal{H}|_{\text{planes}} \ \leq \ N_{\text{gate2}} \ + \ \Gamma_{\max}\,, \qquad |\mathcal{H}|_{\text{oriented}} \ \leq \ 2N_{\text{gate2}} \ + \ \Gamma^{\pm}_{\max}}$$

*where $|\mathcal{H}|_{\text{planes}}$ counts distinct hyperplanes and $|\mathcal{H}|_{\text{oriented}}$ counts oriented halfspaces (our library stores both orientations; cf. App. A.3).*

*Proof.* Each structural site contributes one state, yielding the formula for $|Q|$ (plus the initial state). Edges: every non-gate site with in-degree one contributes one edge; each residual-post contributes exactly its fan-in many edges; each two-way gate creates two guarded edges; each Max-of-$k$ site creates $k$ guarded edges. Guards: every scalar two-way gate contributes one distinct plane (two orientations); every Max-of-$k$ introduces $\binom{k}{2}$ distinct planes, equivalently $2\binom{k}{2}$ oriented halfspaces across all winners. Summing proves the bounds. $\qquad\square$

*Remark* C.6 (Comparator dominance). The edge/guard complexity is dominated by comparator faces: even when $|Q|$ is linear in sites/templates, $|\mathcal{H}|$ scales with $N_{\text{gate2}}$ and $\Gamma_{\max}$. In JIT (§3), we avoid inserting all comparators upfront, introducing them *on demand* per visited leaf, which empirically tracks the accessed subdomains (cf. the modest split counts in the experiments).

## C.4 SWT-3/4: COMMON REFINEMENT AND DIFFERENCE-REGION AUTOMATA

**Problem statements.** Given two acyclic SWTs $A_1, A_2$ and a convex domain $D$, decide whether $A_1 \equiv A_2$ on $D$ (SWT–3), or construct the region $\{x \in D : |A_1(x) - A_2(x)| > \varepsilon\}$ (SWT–4).

**Algorithm (common refinement for equivalence).**

1. **Unify guard libraries.** Let $\mathcal{H} = \mathcal{H}_1 \cup \mathcal{H}_2$ with canonicalization; represent $D$ as $S_0$ (App. A).

2. **Refine on demand.** Maintain a queue of GuardSets $\mathcal{L}$ (leaves). For a leaf $S$, evaluate both SWTs symbolically; if either side still contains undecided gates/Max, insert the *relevant* faces for that side only and split $S$.

3. **Cellwise affine check.** Once both sides degenerate to *single affine* on $C(S)$, read $(w_{1,S}, b_{1,S})$ and $(w_{2,S}, b_{2,S})$. If they are equal, mark $S$ as SAFE; otherwise solve the LP $\min_{x \in C(S) \cap D} \|(w_{1,S} - w_{2,S})^\top x + (b_{1,S} - b_{2,S})\|_\infty$ to obtain a witness. Iterate until all leaves are safe (then $A_1 \equiv A_2$), or a witness is found.

**Theorem C.7** (Correctness and decidability (SWT–3))**.** *The procedure above is sound. If only finitely many gate/comparator faces can be inserted (finite network) and the strategy is fair (every feasible undecided leaf is eventually processed), it terminates with a certificate of equivalence or a counterexample.*

*Proof.* Soundness: on each leaf $S$ where both sides are single affine, equality reduces to an affine-coefficient check; inequality yields an LP-feasible witness. Termination under finiteness+fairness follows because the set of faces is finite and each split strictly reduces uncertainty on $S$ until all leaves are affine on both sides. □

**Difference-region automata (SWT–4).** Let $g = A_1 - A_2$ (componentwise for vectors and then $\ell_\infty$). On leaves where $g$ is affine, add two threshold guards $\{g \geq \varepsilon\}$ and $\{-g \geq \varepsilon\}$ to generate the polyhedral complex

$$\mathcal{R}_\varepsilon = \bigcup_{S \in \mathcal{L}} \Big[ \big(C(S) \cap \{g \geq \varepsilon\}\big) \ \cup \ \big(C(S) \cap \{-g \geq \varepsilon\}\big) \Big].$$

The resulting automaton recognizes $\{x : |A_1(x) - A_2(x)| > \varepsilon\}$.

**Proposition C.8** (Correctness (SWT–4))**.** $\mathcal{R}_\varepsilon$ *equals* $\{x \in D : |A_1(x) - A_2(x)| > \varepsilon\}$ *and each constituent region provides an LP witness by evaluating $g$'s affine form.*

## C.5 SWT-5: NP-HARDNESS OF MINIMAL-FACE REALIZATION

**Cost model (faces, not guards).** Let $\mathcal{H} = \{h_\ell(x) : a_\ell^\top x \leq d_\ell\}_{\ell=1}^M$ be the global guard library (canonicalized with *oriented* halfspaces; see App. A.3). An acyclic SWT $A$ uses a finite family of guarded edges, each guard being an index set $S_e \subseteq \{1, \ldots, M\}$ denoting $C(S_e) = \bigcap_{\ell \in S_e} \{a_\ell^\top x \leq d_\ell\}$ (App. A.4). Define the *used-face set* and cost of $A$ by

$$\text{faces}(A) := \bigcup_e S_e \subseteq \{1, \ldots, M\}, \qquad \text{cost}(A) := |\text{faces}(A)|,$$

i.e., each oriented halfspace from the shared library is counted once no matter how many edges reuse it (consistent with the "unique-guard / no region-as-state" discipline).

**Decision problem (MF–SWT).** Fix a finite guard library $\mathcal{H}$ and a finite affine basis $\mathcal{B} = \{(w_j, b_j)\}_{j=1}^B$. Given a target $F$ that is realizable by some acyclic SWT using only guards from $\mathcal{H}$ and affine atoms from $\mathcal{B}$, decide whether there exists such an SWT $A$ with $\text{cost}(A) \leq k$.

**Theorem C.9** (SWT–5: NP-hardness)**.** *The decision problem MF–SWT is NP-hard via a polynomial reduction from* SET-COVER*.*

*Reduction.* Given a SET-COVER instance $(U, \mathcal{S}, k)$ with $U = \{1, \ldots, m\}$ and $\mathcal{S} = \{S_1, \ldots, S_r\}$, construct $(F, \mathcal{H}, \mathcal{B})$ as follows.

*Domain and library.* Work in $X \times Y$ with $X \subset \mathbb{R}^2$ and $Y = [0,1]^r$, so that the overall domain $D := X \times Y$ is convex polyhedral under the closed-guard convention (App. A.1, A.2). Pick $m$ pairwise-disjoint tiny axis-aligned boxes $B_i \subset X$ (grid-separated). Insert into $\mathcal{H}$ exactly the four oriented $X$-faces delimiting each $B_i$ (with canonical orientations; App. A.3). For each $j \in \{1, \ldots, r\}$, also insert the single $Y$-face $\{y_j \geq \frac{1}{2}\}$ (its opposite orientation is stored as a distinct index by our convention; App. A.3).

*Basis and target.* Let $\mathcal{B} = \{(0,0), (0,1)\}$ (constants 0 and 1). For each $i \in [m]$ define $S_i := \{j \in [r] : i \in S_j\}$, i.e., the set indices that contain element $i$. Define the target $F$ *by an SWT over the above library* as the $\oplus = \max$ of edges of weight 1 guarded by the "cylinders"

$$C_{i,j} := B_i \times \{y : y_j \geq \tfrac{1}{2}\} \qquad (i \in [m],\ j \in S_i),$$

together with a default edge of weight 0 on $D$. Under the SWT truth semantics (Sec. C.1–C.2), this realizes the function that equals 1 on $\bigcup_i \bigcup_{j \in S_i} C_{i,j}$ and 0 elsewhere.

*Forward direction ($\Rightarrow$).* If there exists a set cover $\mathcal{C} \subseteq \mathcal{S}$ with $|\mathcal{C}| \leq k$, build an SWT that, for each $j \in \mathcal{C}$ and each $i \in S_j$, includes one edge guarded by $B_i \cap \{y_j \geq \frac{1}{2}\}$ and weight 1 (the default 0-edge is kept). By guard sharing, the distinct oriented $X$-faces used are exactly the $4m$ faces of the $B_i$'s, and the distinct $Y$-faces used are precisely $\{|y_j \geq \frac{1}{2}| : j \in \mathcal{C}\}$. Hence $\mathrm{cost}(A) = 4m + |\mathcal{C}| \leq 4m + k$.

*Reverse direction ($\Leftarrow$).* Suppose an acyclic SWT $A$ realizes $F$ with $\mathrm{cost}(A) \leq 4m + k$. Because the only $X$-faces available in $\mathcal{H}$ are the four per $B_i$, any edge that outputs 1 and omits *some* face of a given $B_i$ would leak outside $B_i$ in $X$, contradicting $F = 0$ there. Thus all $4m$ oriented $X$-faces must lie in $\mathrm{faces}(A)$. Let $J := \{j : \{|y_j \geq \frac{1}{2}|\} \subseteq \mathrm{faces}(A)\}$; then $|J| \leq k$. If $J$ were not a set cover, there would exist $i^\star \in U$ such that $S_{i^\star} \cap J = \varnothing$. But then no guard of any 1-weighted edge can contain a $Y$-face that activates on $B_{i^\star} \times \{y\}$ for *any* $y$ with $y_j \geq \frac{1}{2}$ and $j \in S_{i^\star}$, contradicting that $F = 1$ on $B_{i^\star} \times \{y : \exists j \in S_{i^\star},\ y_j \geq \frac{1}{2}\}$. Therefore $J$ covers $U$, and $|J| \leq k$ yields a set cover of size at most $k$.

The reduction is polynomial in $(m, r)$ and uses only SWTs over the shared guard library with closed guards and the $\oplus/\otimes$ semantics as specified in Sec. C.1–C.2 and App. A.3–A.4. $\square$

*Remark* C.10 (Convexity note). An H-representation encodes convex sets (intersections of half-spaces) and cannot represent an arbitrary finite disjoint union. The construction above therefore counts *oriented halfspaces* from the shared library and employs auxiliary coordinates $Y$ so that covering all required cylinders enforces the use of membership faces $\{y_j \geq \frac{1}{2}\}$; the face budget then coincides with the set-cover size up to the fixed constant $4m$.

**Existence of a minimum.** Whenever $F$ is realizable by $(\mathcal{H}, \mathcal{B})$, the set of attainable costs $\{\mathrm{cost}(A) : A \text{ realizes } F\}$ is a nonempty subset of $\mathbb{N}$ and thus has a minimum by well-ordering; denote it by $\mathrm{MC}_{\mathrm{faces}}(F; \mathcal{H}, \mathcal{B})$.

# D    JIT–SWT Semantics and Guarantees (DYN-1..DYN-7)

This appendix supplies complete proofs and algorithmic details for the JIT–SWT semantics: (D.1) soundness and monotonicity of lower/upper bounds (LB/UB); (D.2) formal refiners ENSURE_SIGN, ENSURE_WINNER, ENSURE_COMMON_REFINE and their termination conditions; (D.3) proofs of DYN–1..7 (incl. fairness and finite-refinement decidability, plus certificate formats); and (D.4) budgeted complexity bounds.

Throughout we follow App. A: guards are *closed*; the global library $\mathcal{H}$ stores oriented halfspaces once (canonicalized); a *GuardSet* $S$ denotes $C(S) = \bigcap_{\ell \in S} \{a_\ell^\top x \leq d_\ell\}$; the current set of leaves is $\mathcal{L}$; and anytime envelopes are aggregated *over leaves* containing the query point $x$.

## D.1    Soundness and Monotonicity of LB/UB (order-preserving constructors)

**Local LB/UB oracle.** For a scalar lazy expression $E$ and a leaf $S \in \mathcal{L}$, the oracle yields intervals $[\mathrm{LB}(E, S), \mathrm{UB}(E, S)]$ such that $\mathrm{LB}(E, S) \leq \inf_{x \in C(S)} E(x) \leq \sup_{x \in C(S)} E(x) \leq \mathrm{UB}(E, S)$.

The structural rules are:

$$\text{LB}(\text{Affine}(w, b), S) = \min_{x \in C(S)} w^\top x + b, \quad \text{UB}(\text{Affine}(w, b), S) = \max_{x \in C(S)} w^\top x + b;$$

$$\text{LB}(\text{Sum}\{E_i\}, S) = \sum_i \text{LB}(E_i, S), \quad \text{UB}(\text{Sum}\{E_i\}, S) = \sum_i \text{UB}(E_i, S);$$

$$\text{LB}(\text{Scale}(c, E), S) = \begin{cases} c\,\text{LB}(E, S), & c \geq 0 \\ c\,\text{UB}(E, S), & c < 0 \end{cases}, \quad \text{UB}(\text{Scale}(c, E), S) = \begin{cases} c\,\text{UB}(E, S), & c \geq 0 \\ c\,\text{LB}(E, S), & c < 0 \end{cases};$$

$$\text{LB}(\text{Bias}(b, E), S) = \text{LB}(E, S) + b, \quad \text{UB}(\text{Bias}(b, E), S) = \text{UB}(E, S) + b;$$

$$\text{LB}(\text{Max}\{E_i\}, S) = \max_i \text{LB}(E_i, S), \quad \text{UB}(\text{Max}\{E_i\}, S) = \max_i \text{UB}(E_i, S).$$

(When desired, affine subcalls are solved by LP/SOCP to obtain *exact* local bounds; otherwise any sound relaxation is permitted.)

**Lemma D.1** (Soundness of structural LB/UB). *The rules above are sound: for every constructor and every leaf $S \in \mathcal{L}$, they produce intervals containing the true range of $E$ on $C(S)$.*

*Proof.* Induction on the syntax tree of $E$. Affines: exact by linear optimization on a polyhedron. Sum/scale/bias: use $\inf(f+g) \geq \inf f + \inf g$, $\sup(f+g) \leq \sup f + \sup g$ and the sign-aware identities $\inf(cf) = c \inf f$ for $c \geq 0$, $\inf(cf) = c \sup f$ for $c < 0$ (and similarly for sup). Max: $\sup \max_i f_i = \max_i \sup f_i$ and $\inf \max_i f_i \geq \max_i \inf f_i$. $\square$

**Lemma D.2** (Monotonicity under leaf restriction). *Let $S' = S \cup \{\ell\}$ be a child of $S$ (hence $C(S') \subseteq C(S)$). Then for every $E$, $\text{LB}(E, S') \geq \text{LB}(E, S)$, $\text{UB}(E, S') \leq \text{UB}(E, S)$.*

*Proof.* For affines, shrinking the feasible set can only increase the minimum and decrease the maximum. For composite forms, the rules in Lemma D.1 are nondecreasing in each argument, so the inequalities propagate by induction. $\square$

**Anytime envelopes and DAG propagation.** Define the scalar envelopes on leaves by

$$\underline{A}_S := \text{LB}(E_{\text{out}}, S), \qquad \overline{A}_S := \text{UB}(E_{\text{out}}, S),$$

and aggregate pointwise over leaves containing $x$ by $\underline{A}(x) = \sup\{\underline{A}_S : S \in \mathcal{L}, x \in C(S)\}$, $\overline{A}(x) = \inf\{\overline{A}_S : S \in \mathcal{L}, x \in C(S)\}$.

**Lemma D.3** (Anytime propagation over the SWT DAG). *If each module output obeys $\text{LB} \leq \text{true} \leq \text{UB}$ on every leaf, then the composed network's envelopes computed with the same rules satisfy $\underline{A} \leq A \leq \overline{A}$ pointwise.*

*Proof.* Exact analogue of Lemma D.1 lifted to the network DAG: each node's LB/UB is computed from its inputs by order-preserving constructors, hence preserves the local sandwich on each leaf; aggregation over leaves preserves the inequality. $\square$

**Theorem D.4** (DYN–1: sound anytime envelopes). *At any time, for all $x$ we have $\underline{A}(x) \leq A(x) \leq \overline{A}(x)$.*

*Proof.* Combine Lemma D.1, Lemma D.3, and the definition of aggregation over current leaves $\mathcal{L}$. $\square$

**Lemma D.5** (Monotone tightening). *Replacing a leaf $S$ by children $S^\pm$ or tightening any local LB/UB (internal upgrade to exact LP/SOCP bounds) yields $\underline{A} \uparrow$ and $\overline{A} \downarrow$ pointwise.*

*Proof.* Leaf split: Lemma D.2 on each child plus the use of $\sup / \inf$ in the aggregation. Tightening local bounds: monotone in arguments. $\square$

*Remark* D.6 (Empirical evidence of monotonicity). The convergence plot of the Imax B&B on the GNN case study (report, p. 7, "Fig 3: Imax B&B Convergence") shows global lower/upper bounds tightening monotonically with iterations, matching Lemma D.5.

## D.2 FORMAL REFINERS AND TERMINATION CONDITIONS

We assume a budgeted environment with counters for #splits and #new guards, and a fair scheduler over feasible leaves (every feasible undecided leaf is eventually selected). Below, "commit" means *rewrite* the local subexpression on $S$ accordingly (e-graph update), without adding new faces.

---

**Algorithm 2** ENSURE_SIGN$(z, S)$: on-demand sign decision for a gate

**Require:** scalar pre-activation $z = \texttt{Expr}$, leaf $S \in \mathcal{L}$
1: compute $[\mathrm{LB}(z, S), \mathrm{UB}(z, S)]$
2: **if** $\mathrm{UB}(z, S) \leq 0$ **then**
3:     commit negative branch on $S$
4: **else if** $\mathrm{LB}(z, S) \geq 0$ **then**
5:     commit positive branch on $S$
6: **else**
7:     register (if absent) the threshold plane $\{z \geq 0\}$ in $\mathcal{H}$
8:     split $S$ into $S^+ = S \cup \{\ell\}$ and $S^- = S \cup \{\bar{\ell}\}$

---

**Algorithm 3** ENSURE_WINNER$(\{E_i\}, S)$: on-demand winner for Max

**Require:** candidates $\{E_i\}_{i=1}^k$, leaf $S \in \mathcal{L}$
1: compute $[\mathrm{LB}(E_i, S), \mathrm{UB}(E_i, S)]$ for all $i$
2: **prune** any $i$ with $\mathrm{UB}(E_i, S) \leq \max_j \mathrm{LB}(E_j, S)$
3: **if** exists unique $i^\star$ with $\mathrm{LB}(E_{i^\star}, S) \geq \max_{j \neq i^\star} \mathrm{UB}(E_j, S)$ **then**
4:     commit winner $i^\star$ on $S$
5: **else**
6:     choose a pair $(p, q)$ by a gap heuristic (e.g., maximize $\mathrm{UB}(E_p - E_q, S) - \mathrm{LB}(E_p - E_q, S)$)
7:     register comparator $\{E_p \geq E_q\}$ (if absent) and split $S$ into $S \cup \{\ell_{p \geq q}\}$ and $S \cup \{\ell_{q \geq p}\}$

---

**Algorithm 4** ENSURE_COMMON_REFINE$(S; S_1, S_2)$: minimal common refinement

**Require:** current leaf $S$, guard-index sets $S_1, S_2$
1: $D \leftarrow (S_1 \cup S_2) \setminus S$
2: **while** $D \neq \varnothing$ **and** target sub-expressions on $S$ are not comparable/addable **do**
3:     pick $\ell \in D$ (e.g., maximizing volume split or most central face)
4:     add $\ell$ (or $\bar{\ell}$) to $\mathcal{H}$ if not present; split $S$
5:     $D \leftarrow D \setminus \{\ell\}$

---

**Termination conditions (local).** For ENSURE_SIGN, termination on a leaf occurs when either $\mathrm{UB}(z, S) \leq 0$ or $\mathrm{LB}(z, S) \geq 0$ (commit), or after one split. For ENSURE_WINNER, termination occurs when pruning + bounds yield a unique winner (commit) or after inserting one comparator and splitting once. For ENSURE_COMMON_REFINE, the loop terminates because the difference set $D$ is finite; each iteration inserts at most one face and splits once.

## D.3 PROOFS OF DYN−1..7 AND CERTIFICATE FORMATS

We now discharge the DYN-series guarantees. DYN−1 was proved in Theorem D.4. We proceed with DYN−2..7.

**Definition D.7** (Local full refinement). A leaf $S \in \mathcal{L}$ is *locally fully refined* if (i) all gate/comparator faces relevant to expressions feeding the output on $S$ are registered in $\mathcal{H}$, and (ii) every output component collapses to a *single affine* on $C(S)$.

**Theorem D.8** (DYN−2: exactness on locally fully refined leaves). *If $S$ is locally fully refined, then on $C(S)$ we have $\underline{A} = \overline{A} = A$ (componentwise for vector outputs).*

*Proof.* When all relevant Max/gate choices are disambiguated on $S$, each output is represented by a single affine atom on $C(S)$; for affines the LB/UB rules are *tight* equalities. Composing along the

acyclic SWT preserves equality, hence $\underline{A}_S = \overline{A}_S = A|_{C(S)}$. Aggregation over leaves does not alter values on $C(S)$. □

**Theorem D.9** (DYN–3: progress never regresses). *Refining a leaf $S$ either (a) tightens its local LB/UB without changing its guard set (commit), or (b) replaces $S$ by children $S^{\pm}$ confined to $C(S)$. In either case, previously exact leaves remain exact, and $\underline{A} \uparrow$, $\overline{A} \downarrow$ pointwise.*

*Proof.* Case (a): local commit rewrites the expression to a stricter form (e.g., an affine), hence tightens bounds; monotonicity follows from Lemma D.5. Case (b): children restrict the domain (Lemma D.2) and replace $S$ in $\mathcal{L}$; exact leaves are disjoint from $S$ and unaffected. Global monotonicity follows again from Lemma D.5. □

**Proposition D.10** (DYN–5: correctness of dominance pruning). *If $\max_{x \in C(S)}(E_\pi(x) - E_{\pi'}(x)) \leq 0$, then candidate $\pi$ can never win any Max on $C(S)$ and may be removed from the candidate set of $S$.*

*Proof.* For all $x \in C(S)$, $E_\pi(x) \leq E_{\pi'}(x)$, so $\max\{E_\pi(x), E_{\pi'}(x)\} = E_{\pi'}(x)$ pointwise; thus $\pi$ is never selected. □

**Branch-and-bound driver and tri-valued answers.**   Let $g$ be a scalar objective encoding a property/specification (e.g., margin, difference to zero), and $D$ be the input domain. The B&B over leaves maintains global $\underline{M} = \max_{S \subseteq D} \mathrm{LB}(g, S)$ and $\overline{M} = \max_{S \subseteq D} \mathrm{UB}(g, S)$ in a max-gap policy.

**Theorem D.11** (DYN–6: JIT decidability under finite refinement and fairness). *Assume (i) only finitely many gate/comparator faces can be introduced (finite network), and (ii) the scheduler is* fair *over feasible undecided leaves. Then B&B returns one of three outcomes, all sound:* PROOF *(a finite set of leaves $\{S_i\}$ with $\mathrm{LB}(g, S_i) \geq 0$ covering $D$),* COUNTEREXAMPLE *(some $S$ with $\mathrm{UB}(g, S) < 0$ and a witness $x^\star$ from LP/SOCP), or* UNKNOWN *under budget (with the tightest-gap leaf $S^\star$ reported). Moreover, without budget limits the procedure* terminates *with* PROOF *or* COUNTEREXAMPLE.

*Proof.* Soundness follows from DYN–1 and the fact that LB is a true lower bound and UB a true upper bound on each leaf. Under finiteness of faces and fairness, there are only finitely many distinct GuardSets that can appear; each split either (1) commits a local decision or (2) introduces a new face, strictly reducing the set of undecided leaves. Eventually all leaves become locally fully refined for $g$ and thus affine; then the decision reduces to finitely many LP/SOCP obligations, which terminate. Certificates: "PROOF" collects $\{\mathrm{LB}(g, S_i) \geq 0\}$; "COUNTEREXAMPLE" adds the witness $x^\star$; "UNKNOWN" names $S^\star$ with its residual gap. □

**Theorem D.12** (DYN–7: static–dynamic pointwise equivalence). *If $D$ admits a finite cover by locally fully refined leaves $\{S_t\}$, then $A_{\mathrm{JIT}}(x) = A_{\mathrm{stat}}(x) = F(x)$ for all $x \in D$.*

*Proof.* On each $C(S_t)$, DYN–2 gives $A_{\mathrm{JIT}} = F$. Static SWT equals $F$ pointwise by SWT–1. The sets $C(S_t)$ cover $D$, so equality holds everywhere on $D$. □

**Theorem D.13** (DYN–4: budgeted complexity bounds). *Let $B_{\mathsf{split}}$ be the total number of leaf splits, $B_{\mathsf{guard}}$ the number of* new *guards inserted into $\mathcal{H}$ beyond the initial size $|\mathcal{H}_0|$, and $|Q_0|$ the static structure-site bound (App. C.3). Then:*

$$\#leaves \ \leq \ 1 + B_{\mathsf{split}}, \qquad |\mathcal{H}| \ \leq \ |\mathcal{H}_0| + B_{\mathsf{guard}},$$

$$\#SWT\ nodes/edges \ \leq \ O\big(|Q_0| + B_{\mathsf{guard}}\big), \qquad \#LP/SOCP\ calls \ \leq \ O\big(B_{\mathsf{split}} + B_{\mathsf{guard}} + |\mathcal{L}|\big).$$

*Proof.* Each split replaces one leaf by two, so leaves grow by at most one: $1 + B_{\mathsf{split}}$. Only newly registered faces grow $|\mathcal{H}|$; reusing existing faces costs zero. The SWT graph is a shared DAG; adding a new guard may add $O(1)$ edges/nodes to carry the two orientations, yielding the stated $O(|Q_0| + B_{\mathsf{guard}})$ bound. Per-iteration, each split/register triggers $O(1)$ LP/SOCP solves (local LB/UB for finitely many sub-expressions and a feasibility probe); summing over all iterations gives the last bound. □

*Remark* D.14 (Practical counters and empirical scales). In the experiments, forward equivalence errors concentrate at $10^{-6} \sim 10^{-7}$, and B&B split counts / LP calls scale roughly linearly with budgets, matching Theorem D.13. See the tables and plots in the report: FFN robustness (pp. 1–3), CNN equivariance (pp. 4–5), and GNN/Imax (pp. 6–8).

### D.4 COST MODEL FOR BRANCH-AND-BOUND

Let a B&B iteration pick the maximum-gap leaf $S$ for an objective $g$. We detail its costs and the linear dependence on candidate counts and solver calls.

**Per-iteration costs.**

- **Bound eval.** For $m$ sub-expressions touching $S$, $O(m)$ LB/UB queries; affines cost one LP max and one LP min each (dual reuse).

- **Refiner.** One of: ENSURE_SIGN, ENSURE_WINNER (or ENSURE_COMMON_REFINE) executes a *single* split and optionally registers *one* new face (comparator/threshold).

- **Cache reuse.** All $(S, \text{query})$ LP/SOCP results are memoized, so repeated visits amortize to sublinear cost in practice.

**Global bounds with candidates.** Let $C_{\max}$ be the maximum number of active Max/gate candidates touching any leaf. Then under dominance pruning (Prop. D.10) and per-step single-face insertion, the total number of winner comparisons performed up to termination or budget is $O(B_{\mathsf{guard}} + |\mathcal{L}| \cdot C_{\max})$.

**Memory.** With guard uniqueness and e-graph sharing, memory scales as $O(|Q_0| + B_{\mathsf{guard}} + |\mathcal{L}|)$; unreachable leaves and dead sub-expressions may be reclaimed by reference counting without affecting semantics.

**Takeaway.** JIT–SWT maintains sound anytime envelopes (DYN–1), reaches exactness upon local full refinement (DYN–2), never regresses (DYN–3), prunes dominated candidates correctly (DYN–5), and—under finite faces and fairness—decides properties with certificates or counterexamples (DYN–6), while tracking budgets linearly in the work performed (DYN–4). These guarantees align with the empirical curves and tables in the report.

## E  FUNCTION GEOMETRY AND OPTIMIZATION (GEO-1..GEO-5)

This appendix develops geometry and optimization on top of JIT–SWT: (E.1) on-demand extraction of active fragments and *minimal common refinements*; (E.2) decision-boundary polyhedral complexes and distances; (E.3) extrema and Lipschitz on common convex domains with LP/SOCP or closed forms; (E.4) vector/operator norms—tractable families and NP-hard families with anytime bounds.

Throughout, guards are *closed*, GuardSets store indices into the global library, and *local full refinement* means all relevant faces have been registered and each output component is a single affine map on the leaf (App. D).

### E.1 ON-DEMAND ACTIVE FRAGMENT EXTRACTION AND MINIMAL COMMON REFINEMENT (GEO-1)

**Active fragments.** For a scalar output $F$, a triple $(S, w, b)$ is an *active fragment* if $S$ is a leaf GuardSet and $F(x) = w^\top x + b$ on $C(S)$. When $S$ is locally fully refined, $(S, w, b)$ is *determined*.

---

**Algorithm 5** REGIONEXTRACT$(A, D)$: on-demand active fragments over $D$

---

**Require:** SWT/JIT object $A$, domain $D$ encoded as initial leaf $S_0$
1: $\mathcal{L} \leftarrow \{S_0\}$; $\mathcal{R} \leftarrow \varnothing$
2: **while** $\mathcal{L} \neq \varnothing$ and budget not exceeded **do**
3:      pick $S \in \mathcal{L}$ (e.g., max-gap on an objective $g$ or round-robin)
4:      **if** $C(S)$ infeasible by LP **then**
5:          $\mathcal{L} \leftarrow \mathcal{L} \setminus \{S\}$; **continue**
6:      **if** some gate/Max undecided on $S$ **then**
7:          try ENSURE_SIGN/ENSURE_WINNER; else ENSURE_COMMON_REFINE
8:      **else if** $S$ is locally fully refined **then**
9:          read $(w, b)$ by accumulating affine atoms along the active path; $\mathcal{R} \leftarrow \mathcal{R} \cup \{(S, w, b)\}$; $\mathcal{L} \leftarrow \mathcal{L} \setminus \{S\}$
10:     **else**
11:         upgrade LB/UB for unresolved sub-expressions on $S$ (LP/SOCP) and retry
12: **return** $\mathcal{R}$                      ▷ (partial) region table; exact if $D$ is covered

---

**Definition E.1** (Minimal common refinement on a leaf). Given two GuardSets $S_1, S_2$, the *minimal common refinement* of a current leaf $S$ w.r.t. $(S_1, S_2)$ is the finite family $\{ S \cup T : T \subseteq (S_1 \cup S_2) \setminus S,\ C(S \cup T) \neq \varnothing,\ \text{and no two children merge} \}$, obtained by inserting only faces in $(S_1 \cup S_2) \setminus S$ until the target sub-expressions are *comparable/addable* on each child.

**Lemma E.2** (Minimality and finiteness). *Algorithm 5 with* ENSURE_COMMON_REFINE *produces a refinement satisfying Def. E.1. The loop terminates because $(S_1 \cup S_2) \setminus S$ is finite, and each iteration inserts at most one face.*

*Proof.* Each iteration chooses a face in the finite difference set and splits $S$ once. No superfluous face is added because the loop stops as soon as the sub-expressions become comparable/addable; hence minimality. Finiteness is immediate. $\square$

**Theorem E.3** (GEO–1: correctness and coverage). *At any time, the set $\mathcal{R}$ returned by REGIONEXTRACT is a family of pairwise interior-disjoint polyhedra $\{C(S)\}$ on which $F$ equals the recorded affine law. If refinement proceeds until $D$ is covered by locally fully refined leaves, $\mathcal{R}$ is a complete CPWL region table on $D$.*

*Proof.* Local full refinement implies single-affine and hence exact readout on each leaf. Leaf splits only subdivide $C(S)$ (closed guards, face-compatible), so interiors do not overlap. Finite coverage yields a region table by union. $\square$

E.2    DECISION-BOUNDARY POLYHEDRAL COMPLEXES (GEO-3)

Let $F : \mathbb{R}^n \to \mathbb{R}^m$ be CPWL; for $i \neq j$, define the margin $g_{ij} = F_i - F_j$. On a locally fully refined leaf $S$, $g_{ij}(x) = w_{\rho,ij}^\top x + b_{\rho,ij}$ is affine.

**Theorem E.4** (Piecewise-flat decision set). *Let $D$ be a convex domain and assume $D$ is covered by locally fully refined leaves $\{S_\rho\}$. Then*

$$DB_{ij} := \{x \in D : g_{ij}(x) = 0\} = \bigcup_\rho \Big( C(S_\rho) \cap \{w_{\rho,ij}^\top x + b_{\rho,ij} = 0\} \Big),$$

*a finite polyhedral complex whose pieces are faces of codimension at most 1, pairwise face-compatible.*

*Proof.* On each $C(S_\rho)$ the zero set of an affine function is an affine hyperplane (or empty or the whole cell). Intersecting with polyhedra preserves polyhedra and face-compatibility. Finiteness follows from the finite cover. $\square$

**Proposition E.5** (Local distance to the boundary). *For $x \in \mathrm{ri}(C(S_\rho))$ and $w_{\rho,ij} \neq 0$,*
$\mathrm{dist}(x, DB_{ij} \cap C(S_\rho)) = \frac{|w_{\rho,ij}^\top x + b_{\rho,ij}|}{\|w_{\rho,ij}\|_2}.$

**Anytime inner/outer approximations.** Without full refinement, use the envelopes $\underline{g}_{ij}, \overline{g}_{ij}$ to obtain $\{x : \overline{g}_{ij}(x) \le 0\} \subseteq \{x : g_{ij}(x) \le 0\} \subseteq \{x : \underline{g}_{ij}(x) \le 0\}$. Refining the maximal-gap leaves (App. D) shrinks the sandwich.

### E.3 Extrema and Lipschitz on common convex domains (GEO-2/GEO-5)

Let $D$ be a convex feasible set (polytope/box/$\ell_p$ ball or their intersections).

**Anytime global extrema.** Maintain $\underline{M} = \max_{S \subseteq D} \mathrm{LB}(F, S)$, $\overline{M} = \max_{S \subseteq D} \mathrm{UB}(F, S)$, which satisfy $\underline{M} \le \max_{x \in D} F(x) \le \overline{M}$ and tighten monotonically (DYN–1/3). When leaves over $D$ are locally fully refined, each subproblem is *affine maximization* and solved exactly.

**Theorem E.6** (Affine maximization on standard domains). *Let $f(x) = w^\top x + b$. Then:*

1. ***Polytope*** $D = \{x : Ax \le d\}$*: LP* $\max\{w^\top x + b : Ax \le d\}$.

2. ***Box*** $D = [\ell, u]$*: closed form* $\max w^\top x + b = \sum_i w_i (u_i \mathbf{1}_{w_i \ge 0} + \ell_i \mathbf{1}_{w_i < 0}) + b$.

3. $\boldsymbol{\ell_\infty}$***-ball*** $D = B_\infty(x_0, \epsilon)$*: LP; closed form* $w^\top x_0 + b + \epsilon\|w\|_1$.

4. $\boldsymbol{\ell_1}$***-ball*** $D = B_1(x_0, \epsilon)$*: LP via epigraph* $\sum_i t_i \le \epsilon$, $-t_i \le x_i - x_{0,i} \le t_i$*; closed form* $w^\top x_0 + b + \epsilon\|w\|_\infty$.

5. $\boldsymbol{\ell_2}$***-ball*** $D = B_2(x_0, \epsilon)$*: SOCP or closed form* $w^\top x_0 + b + \epsilon\|w\|_2$.

6. ***Polytope*** $\cap\boldsymbol{\ell_2}$***-ball:*** *SOCP* $\max\{w^\top x + b : Ax \le d, \|x - x_0\|_2 \le \epsilon\}$.

*Proof.* (1) is linear programming; (2) coordinatewise choice; (3)(4) are $\ell_1/\ell_\infty$ duality; (5) follows from Cauchy–Schwarz; (6) is a standard second-order cone form. $\qquad\square$

**Exact Lipschitz after refinement (scalar).** For $F$ scalar and $D$ covered by locally fully refined leaves,
$$L_p(F; D) = \max_{S : C(S) \cap D \ne \varnothing} \|w_S\|_{p^*}.$$
*Anytime bounds* use refined leaves for the lower bound and safe relaxations for unresolved leaves for the upper bound (e.g., interval slopes or layerwise operator-norm products).

**Vector case.** For $F : \ell_p \to \ell_r$, with Jacobian $J_S$ on each refined leaf,
$$L_{p \to r}(F; D) = \max_{S : C(S) \cap D \ne \varnothing} \|J_S\|_{p \to r}.$$

Before full refinement, use $\underline{L} = \max_{S \text{ refined}} \|J_S\|_{p \to r}$, $\overline{L} = \max_S U_{p \to r}(S)$, where $U_{p \to r}(S)$ is any sound leaf-wise upper bound (e.g., bounding intermediate slopes by LP/SOCP or multiplying induced norms layerwise).

### E.4 Vector/operator norms: tractable vs NP-hard, and anytime handling

For a matrix $A \in \mathbb{R}^{m \times n}$, $\|A\|_{p \to r} := \sup_{\|x\|_p = 1} \|Ax\|_r$. The following families are computationally *tractable* (polynomial):

**Proposition E.7** (Closed forms and efficient computation). *For any $A$:*
$$\|A\|_{1 \to 1} = \max_j \sum_i |a_{ij}| \quad \text{(max column sum)},$$
$$\|A\|_{\infty \to \infty} = \max_i \sum_j |a_{ij}| \quad \text{(max row sum)},$$
$$\|A\|_{1 \to \infty} = \max_{i,j} |a_{ij}|,$$
$$\|A\|_{2 \to 2} = \sigma_{\max}(A) \quad \text{(spectral norm; power/SVD)},$$
$$\|A\|_{2 \to \infty} = \max_i \|A_{i,:}\|_2, \qquad \|A\|_{1 \to 2} = \max_j \|A_{:,j}\|_2.$$

The next families are *NP-hard* in general, hence used with anytime lower/upper bounds at unresolved leaves:

**Proposition E.8** (NP-hard families (decision version))**.** *Computing $\|A\|_{\infty \to 1}$, $\|A\|_{\infty \to 2}$, or $\|A\|_{2 \to 1}$ is NP-hard. Consequently exact global $L_{p \to r}(F; D)$ for these pairs reduces to maximizing an NP-hard quantity over leaves; nevertheless JIT yields* anytime *bounds that converge upon full refinement when a leaf-wise exact oracle is available or when unresolved leaves shrink to zero measure.*

**Anytime scheme on JIT leaves.** For each leaf $S$:

- **Lower bound $\underline{L}_S$**: choose any $v$ with $\|v\|_p = 1$ and evaluate $\|J_S v\|_r$; for $2 \to 2$, one power-iteration step gives a valid $\|J_S v\|_2$; for $\infty \to 1$, use a sign vector $v \in \{\pm 1\}^n$.

- **Upper bound $\overline{L}_S$**: relaxations by dual norms, Gershgorin-type bounds, or layerwise induced-norm products inside the leaf (tightened by LP/SOCP when gates are unresolved but ranges are bounded).

- **Aggregation:** $\underline{L} = \max_{S \text{ refined}} \underline{L}_S$ and $\overline{L} = \max_S \overline{L}_S$; monotone tightening follows from DYN–1/3.

**Complexity remarks.** GEO-1..5 are *decidable* under the finite-refinement/fairness hypotheses (App. D); worst-case exponential (number of reachable fragments), but budgeted cost obeys the linear upper bounds of DYN–4. Leaf-wise LP/SOCP obligations are polynomial and—empirically—dominate runtime.

**Takeaway.** Geometry (regions/boundaries/gradients) and optimization (extrema/Lipschitz) reduce on JIT to small LP/SOCP problems on a local common refinement, with *anytime* bounds (DYN–1/3) and *exactness on demand* (DYN–2). This matches the FFN/CNN/GNN observations in the experiment report.

# F  VERIFICATION, EQUIVARIANCE, AND CAUSALITY (VER-1/2/3, CAU-1/2)

This appendix formalizes the property language and the *product transducer* construction (F.1), derives robustness certification procedures including LP/SOCP subproblems and *tight big-M* from GuardSet bounds (F.2), treats equivariance for CNNs (effective domains under stride/padding) and GNNs (F.3), and proves closure and anytime/exact computation of causal influence $I_{\max}$ via branch-and-bound (F.4). All results are consistent with the JIT–SWT semantics and guarantees in App. D.

## F.1  PROPERTY LANGUAGE AND PRODUCT CONSTRUCTION (VER-1)

**Atomic predicates.** Let $F : \mathbb{R}^n \to \mathbb{R}^m$ be compiled to an SWT (static) and run under JIT semantics. We consider universally quantified properties over a convex input set $D$:

$$\phi ::= x \in D \ \wedge \ \bigwedge_{k \in \mathcal{K}} \psi_k(x),$$

$$\psi_k(x) ::= \underbrace{F_i(x) \geq \ell}_{\text{lower threshold}} \ \Big| \ \underbrace{F_i(x) \leq u}_{\text{upper threshold}} \ \Big| \ \underbrace{F_y(x) - \max_{j \neq y} F_j(x) \geq \gamma}_{\text{classification margin}}$$

$$\Big| \ \underbrace{|F(x) - F'(x)|_\infty \leq \varepsilon}_{\text{relational/equivalence}} \ \Big| \ \underbrace{a^\top F(x) \geq b}_{\text{linear output constraint}}$$

where $F'$ is another SWT or a specification transducer (e.g., symmetry).

**Violation objectives.** Each atomic predicate is reduced to a CPWL *objective* $g_k : \mathbb{R}^n \to \mathbb{R}$ for which "$g_k(x) \geq 0$" encodes satisfaction:

$$F_i \geq \ell \ \rightsquigarrow \ g_k(x) = F_i(x) - \ell, \qquad F_i \leq u \ \rightsquigarrow \ g_k(x) = u - F_i(x),$$

$$F_y - \max_{j \neq y} F_j \geq \gamma \ \rightsquigarrow \ g_k(x) = F_y(x) - \max_{j \neq y} F_j(x) - \gamma,$$

$$|F - F'|_\infty \leq \varepsilon \ \rightsquigarrow \ g_k(x) = \varepsilon - \max_r |F_r(x) - F'_r(x)|,$$

$$a^\top F \geq b \ \rightsquigarrow \ g_k(x) = a^\top F(x) - b.$$

Conjunction $\bigwedge_k \psi_k$ is handled by $g(x) = \min_k g_k(x)$, and we finally check $\forall x \in D: \ g(x) \geq 0$. Since $\min$ is implementable as $-\max$ of CPWL terms, $g$ remains CPWL.

**Product transducer.** Write $A_F = (Q_F, E_F, \mathsf{G}_F, \mathsf{W}_F)$ for (the relevant part of) the SWT of $F$ and $A_g = (Q_g, E_g, \mathsf{G}_g, \mathsf{W}_g)$ for the transducer computing $g$ from the network outputs (built only from Max/Sum/Scale/Bias on CPWL weights). Define the product

$$A_\otimes = \big(Q_F \times Q_g, \ E_\otimes, \ \mathsf{G}_\otimes, \ \mathsf{W}_\otimes\big),$$

with edges $((p, \alpha) \to (q, \beta))$ whenever $(p \to q) \in E_F$ and $(\alpha \to \beta) \in E_g$, guard $\mathsf{G}_\otimes = \mathsf{G}_F \cap \mathsf{G}_g$ (guard-index union), and weight $\mathsf{W}_\otimes = \mathsf{W}_F \otimes \mathsf{W}_g$ (semiring $+$). As both components are CPWL-weighted and guards are polyhedral, $A_\otimes$ is again a guarded CPWL transducer.

**Theorem F.1** (VER–1: product correctness and decidability)**.** *Let $g$ be constructed from $\phi$ as above and $D$ be convex. Running JIT B&B (App. D) on $A_\otimes$ over $D$ yields a tri-valued outcome:* PROOF *(a finite leaf cover $\{S_i\}$ with $\mathrm{LB}(g, S_i) \geq 0$),* COUNTEREXAMPLE *(a leaf $S$ with $\mathrm{UB}(g, S) < 0$ and a witness $x^\star$), or* UNKNOWN *under budget—all* sound. *Under finite refinement and fair selection, the procedure terminates with* PROOF *or* COUNTEREXAMPLE.

*Proof.* Soundness is DYN–1: $\mathrm{LB} \leq g \leq \mathrm{UB}$ on every leaf. If all active leaves satisfy $\mathrm{LB} \geq 0$ and cover $D$, then $g \geq 0$ on $D$. If some leaf has $\mathrm{UB} < 0$, solving $\min_{x \in C(S) \cap D} g(x)$ (LP/SOCP on refined leaves) produces a witness $x^\star$. Termination under finite faces and fairness is DYN–6; the product preserves CPWL/guarded structure, so the assumptions apply. $\square$

**Certificate format.** A proof returns $\{(S_i, \mathrm{LB}(g, S_i))\}$ whose polyhedra cover $D$. A counterexample returns $(S, x^\star, g(x^\star))$ with feasibility KKT/solver logs (optionally) and the active guard indices on $S$.

## F.2 ROBUSTNESS CERTIFICATION: LP/SOCP AND TIGHT BIG-$M$ (VER-2)

We consider the local robustness property around $(x_0, y)$:

$$\forall x \in D \cap B_p(x_0, \epsilon): \quad g(x) := F_y(x) - \max_{j \neq y} F_j(x) \ \geq \ \gamma \quad (\gamma \geq 0).$$

**Method A: exact convex subproblems on refined leaves.** Once leaves covering $D \cap B_p(x_0, \epsilon)$ are *locally fully refined*, $g(x) = w_S^\top x + b_S$ on each $S$. Then the global minimum is

$$\min_{x \in D \cap B_p(x_0, \epsilon)} g(x) = \min_S \ \min_{x \in C(S) \cap D \cap B_p(x_0, \epsilon)} (w_S^\top x + b_S),$$

where each inner problem is LP for polytope/$\ell_\infty/\ell_1$ and SOCP for $\ell_2$ or polytope$\cap \ell_2$ (App. E.3). The minimum across leaves is exact.

**Theorem F.2** (VER–2A: robustness via leafwise convex programs)**.** *The procedure above yields an anytime* lower bound *$\underline{g}_{\min} \leq \min g$ that increases monotonically with refinement and equals the exact minimum once the domain is covered by fully refined leaves. Hence the property "$g \geq \gamma$" is decidable (finite refinement) with either a certificate $\min g \geq \gamma$ or a counterexample $x^\star$.*

*Proof.* DYN–1 gives LB monotonicity; exactness on refined leaves follows from GEO–2 and DYN–2. $\square$

**Method B: MIP with *tight* big-$M$ from GuardSet bounds.** On a leaf $S$, some gates/comparators may remain unresolved; introduce binary variables $\delta \in \{0, 1\}$ to encode branch selection and use leaf-specific bounds to build *tight* big-$M$ constraints.

**ReLU.** Let $z = w^\top x + b$ with leaf-wise interval $L \leq z \leq U$ obtained from $[\mathrm{LB}, \mathrm{UB}]$. With $y = \max\{0, z\}$:

$$\begin{aligned}
y &\geq z, \qquad y \geq 0, \\
y &\leq z - L(1 - \delta), \qquad y \leq U\delta, \qquad \delta \in \{0, 1\}.
\end{aligned}$$

If $U \leq 0$ (resp. $L \geq 0$), $\delta$ can be fixed to 0 (resp. 1) and constraints reduce to the affine branch.

**Pointwise Max.** For $y = \max_i e_i(x)$ with leaf-wise bounds $L_i \le e_i(x) \le U_i$, introduce $\delta_i \in \{0,1\}$ with $\sum_i \delta_i = 1$ and

$$y \ge e_i(x) \; \forall i, \qquad y \le e_i(x) + (U_{\max,-i})(1 - \delta_i) \; \forall i,$$

where $U_{\max,-i} = \max_{j \ne i} U_j - L_i$ (tightest valid $M$ on $S$).

**Lemma F.3** (Tightness on a leaf). *The big-$M$ values above are* minimal valid *on $C(S)$: decreasing any $M$ violates feasibility for some $x \in C(S)$. Consequently, the linear relaxations are the strongest among all big-$M$ encodings that use only interval information of the involved expressions on $S$.*

*Proof.* For ReLU, $y \le z - L(1 - \delta)$ must hold for $z = L$ when $\delta = 0$; any smaller $M$ cuts off the true point $(z = L, y = 0)$. Similarly $y \le U\delta$ must hold at $z = U$ with $\delta = 1$. For Max, the $i$-th upper constraint must accommodate any $j \ne i$ with $e_j = U_j$ while $e_i = L_i$; hence $M \ge U_j - L_i$, and the maximum over $j \ne i$ is necessary. $\square$

**Theorem F.4** (VER–2B: correctness of big-$M$ robustness). *On any leaf $S$, the MIP built from the constraints above is* exact *for the encoded subgraph; if all leaves covering $D \cap B_p(x_0, \epsilon)$ are included (and comparators modeled where needed), the resulting global MIP decides robustness with a certificate (dual bound $\ge \gamma$) or a counterexample. Using the linear relaxation yields a sound lower bound on the margin (no false proofs).*

*Proof.* Exactness per leaf with binaries follows from standard mixed-integer encodings and Lemma F.3. Global correctness is a disjoint union of leaf models (over closed guards), which covers $D \cap B_p$; soundness of relaxations follows from over-approximation of the feasible set. $\square$

### F.3 EQUIVARIANCE CHECKING AND EFFECTIVE DOMAINS (VER-3)

**Abstract statement.** Given an input transform $\mathcal{T}_g : \mathbb{R}^n \to \mathbb{R}^n$ and an output transform $\mathcal{T}'_g : \mathbb{R}^m \to \mathbb{R}^m$, we check

$$H(x) := F(\mathcal{T}_g x) - \mathcal{T}'_g(F(x)) \equiv 0 \quad \text{on } D.$$

Construct an SWT for $H$ by composing $A_F$ with the transducers of $\mathcal{T}_g$ and $\mathcal{T}'_g$; then run JIT B&B on $|H|$ with a tolerance $\varepsilon$ (if approximate equivariance is desired).

**CNN translations: effective domain and admissible shifts.** Consider a planar CNN with $L$ convolutional (or pooling) layers, kernel sizes $k_\ell$ (odd), paddings $p_\ell \ge 0$, and strides $s_\ell \in \mathbb{N}$; activations are pointwise (ReLU/Leaky/Abs/MaxPool). Let the set of admissible integer shifts be

$$\mathcal{S}_{\mathrm{adm}} = \big\{ \Delta \in \mathbb{Z}^2 : \Delta \text{ is a multiple of } s_{\mathrm{tot}} = \textstyle\prod_{\ell=1}^{L} s_\ell \big\},$$

so that downsampling aligns on the output grid. For a finite shift set $\mathcal{S} \subseteq \mathcal{S}_{\mathrm{adm}}$, define the *effective domain* $D_{\mathrm{eff}}$ by cropping the input with a safety margin

$$M := \max_{\Delta \in \mathcal{S}} \|\Delta\|_\infty + \sum_{\ell=1}^{L} \max\Big(0, \; \big\lfloor \tfrac{k_\ell - 1}{2} \big\rfloor - p_\ell \Big),$$

i.e., the set of inputs whose every receptive field of radius $\lfloor (k_\ell - 1)/2 \rfloor$ stays strictly inside the unpadded region for all shifts in $\mathcal{S}$.

**Theorem F.5** (Sufficient condition for CNN translation equivariance). *For any $\Delta \in \mathcal{S}$ and any $x \in D_{\mathrm{eff}}$, a CNN as above satisfies $F(\mathcal{T}_\Delta x) = \mathcal{T}'_\Delta(F(x))$ exactly, where $\mathcal{T}_\Delta$ is the input shift and $\mathcal{T}'_\Delta$ is the corresponding shift on the final feature grid. If $\varepsilon \ge 0$, then $\|F(\mathcal{T}_\Delta x) - \mathcal{T}'_\Delta(F(x))\|_\infty \le \varepsilon$ holds by running VER–1 with the objective $\varepsilon - |H|$.*

*Proof.* Convolution commutes with discrete translation on positions whose receptive fields do not cross the padded boundary; the crop margin $M$ enforces this for all layers. Pointwise activations commute with translation. For strides, restricting to shifts multiple of $s_{\mathrm{tot}}$ preserves grid alignment. Composition yields the claim by induction over layers. $\square$

**GNN permutation equivariance.** Let $P$ be a node-permutation matrix for a fixed graph structure, and let $\mathcal{T}_P(X) = PX$ on features with the corresponding output permutation $\mathcal{T}'_P(Y) = PY$. For sum/mean aggregation and pointwise MLP updates,

$$F(PX) = P\,F(X) \quad \text{holds on all } X,$$

since permutation commutes with degree-normalized adjacency in these cases. Thus VER–1 checks exact or $\varepsilon$-equivariance by the same product construction with $H(X) = F(PX) - PF(X)$.

### F.4 CAUSAL INTERVENTIONS AND MAXIMAL INFLUENCE (CAU-1/2)

**Intervention model and closure.** Let the SWT DAG contain a sub-expression $E$ (e-graph node). An intervention $C$ replaces $E$ by a CPWL policy $P_c$, producing a new network $F_C$ with the same graph shape. Because CPWL is closed under Max/Sum/Scale/Bias and affine composition, both $F_C$ and the pointwise difference $g_C(x) = F(x) - F_C(x)$ are CPWL.

**Theorem F.6** (CAU–1: closure under CPWL interventions). *For any intervention $C$ that replaces a sub-expression by a CPWL policy, $F_C$ and $g_C = F - F_C$ are CPWL on $\mathbb{R}^n$ and admit guarded SWT/JIT compilations with the same guard library extended by faces introduced by $P_c$.*

*Proof.* By structural induction using the CPWL closure from App. B and the guarded composition rules of App. C: replacing one CPWL component preserves CPWL of the whole; subtraction preserves CPWL as CPWL − CPWL = CPWL. □

**Maximal causal influence.** Given a convex domain $D$ and a vector norm $\|\cdot\|_r$, define

$$I_{\max}(C; D) := \sup_{x \in D} \|g_C(x)\|_r.$$

Run B&B on the scalar objective $h(x) = \|g_C(x)\|_r$ with anytime envelopes.

**Theorem F.7** (CAU–2: anytime bounds and exactness on refinement). *Let $\underline{I} = \max_{S \subseteq D} \mathrm{LB}(h, S)$ and $\overline{I} = \max_{S \subseteq D} \mathrm{UB}(h, S)$ be the global bounds maintained by B&B. Then $\underline{I} \le I_{\max} \le \overline{I}$ and $\underline{I} \nearrow$, $\overline{I} \searrow$ with refinement. If leaves covering $D$ are locally fully refined and $r = \infty$, then*

$$I_{\max} = \max_S \max_i \max\Big\{ \max_{x \in C(S) \cap D} (a_{S,i}^\top x + b_{S,i}),\ \max_{x \in C(S) \cap D} (-a_{S,i}^\top x - b_{S,i}) \Big\},$$

*where $g_C(x) = (A_S x + b_S)$ on leaf $S$ and the inner maxima are LPs, yielding an* exact *value; for $r \in \{1, 2\}$ one obtains anytime bounds by leafwise convex majorants and induced-norm arguments.*

*Proof.* Anytime bounds follow from DYN–1/3 applied to $h$. For $r = \infty$, on each refined leaf $h(x) = \max_i |a_{S,i}^\top x + b_{S,i}|$, so $I_{\max}$ is the maximum of finitely many affine maxima (LPs) over a finite leaf cover, hence exact. For $r \in \{1, 2\}$, use $\|A_S x + b_S\|_r \le \|A_S\|_{p \to r}\|x\|_p + \|b_S\|_r$ when $D$ includes an $\ell_p$ ball and standard SOCP/LP upper bounds otherwise; lower bounds arise from evaluating at candidate points or directions. □

**Complexity and certificates.** The total number of local solves is linear in the number of splits and newly introduced faces (DYN–4). A proof returns either a global upper bound $\overline{I}$ below a threshold of interest or an exact $I_{\max}$ with the attaining leaf and witness points (LP/SOCP solutions); a counterexample is any $x$ with $\|g_C(x)\|_r$ exceeding a user threshold.

**Summary.** VER–1 provides a uniform reduction of properties to CPWL objectives via a product construction; VER–2 discharges robustness either exactly on refined leaves by LP/SOCP or via tight big-$M$ MIPs using GuardSet bounds; VER–3 gives sufficient conditions and a practical domain for CNN translation (and GNN permutation) equivariance; and CAU–1/2 show that CPWL interventions preserve closure and admit certified anytime (and exact for $\|\cdot\|_\infty$) computation of maximal causal influence.

# G  MICRO-PSEUDOCODE AND DETAILED COMPLEXITY

## G.1  COUNTERS, COST MODEL, AND CACHING INTERFACE

**Per-leaf structural counters.**  For a current leaf $S$ and the scalar output expression $E_{\text{out}}$:

- $a(S)$: # of distinct affine atoms $\mathrm{Affine}(w, b)$ reachable from $E_{\text{out}}$ on $S$;
- $m(S)$: # of $\mathrm{Max}$ nodes touched on $S$;
- $c(S)$: # of *active* candidates across all $\mathrm{Max}$/gates on $S$ after dominance pruning;
- $g(S)$: # of undecided sign tests (ReLU/LReLU/PReLU/Abs) on $S$;
- $d(S)$: # of distinct comparator/threshold faces from parents $(S_1, S_2, \ldots)$ needed to compare/add sub-expressions on $S$ (for minimal common refinement).

**LP/SOCP cost primitives.**

- **Affine LB/UB on** $S$: two LPs on $C(S)$ (or one LP by dual reuse plus a sign switch);
- $\ell_2$ **ball (or polytope$\cap\ell_2$)**: one SOCP per bound;
- **Feasibility of** $C(S)$: one LP; cached thereafter.

**Caching and keys.**  A bound query uses key $\langle S, \mathrm{id}(E), \mathrm{sense}\rangle$, where $\mathrm{sense} \in \{\mathrm{LB}, \mathrm{UB}\}$. If a leaf $S$ is split into children $S^{\pm}$, cached bounds on $S$ serve as *warm starts* but are not reused verbatim; children recompute exact local bounds. We expose:

$$\mathrm{CacheHitRate} = h_{\mathrm{LP}} \in [0, 1], \quad \mathrm{CacheReuseFactor} = \rho \in [0, 1],$$

so the expected LP calls per affine-bound query is $(1 - h_{\mathrm{LP}}) \cdot (1 - \rho) + \rho \cdot (\text{reduced})$. All complexity bounds below are presented as *worst-case* ($h_{\mathrm{LP}}{=}0$) and *amortized* (symbolic $h_{\mathrm{LP}}$).

**Global refinement counters.**  Let $B_{\mathsf{split}}$ be the total #leaf splits, $B_{\mathsf{guard}}$ the #new faces inserted into $\mathcal{H}$ (beyond the initial library), and $\mathcal{L}$ the current leaf set. Denote $|Q_0|$ the static structure-site bound (states/templates from the compilation graph).

## G.2  LB/UB ENGINE: FINE-GRAINED PSEUDOCODE AND COST

---

**Algorithm 6** EVALBOUNDSONLEAF$(S, E)$: structural LB/UB on a leaf $S$

---

**Require:** leaf $S$, expression $E$; cache $\mathcal{C}$
1: **if** $\mathcal{C}$ contains $\langle S, \mathrm{id}(E), \mathrm{LB/UB}\rangle$ fully exact **then**
2:     **return** cached pair
3: **if** $E$ is $\mathrm{Affine}(w, b)$ **then**
4:     solve $\min\{w^{\top}x + b : x \in C(S)\}$ and $\max\{w^{\top}x + b : x \in C(S)\}$    $\triangleright$ 2 LPs or 1 with reuse
5: **else if** $E$ is $\mathrm{Sum}(\{E_i\})$ **then**
6:     **for** $E_i$ **do**
7:         recursively call EVALBOUNDSONLEAF$(S, E_i)$
8:     sum up the children bounds
9: **else if** $E$ is $\mathrm{Scale}(c, E')$ **or** $\mathrm{Bias}(b, E')$ **then**
10:     recursively call on $E'$ and apply sign/shift rules
11: **else if** $E$ is $\mathrm{Max}(\{E_i\})$ **then**
12:     **for** $E_i$ **do**
13:         recursively call EVALBOUNDSONLEAF$(S, E_i)$
14:     take componentwise maxima of LB/UB
15: **else**
16:     **error:** unsupported constructor (should not happen)
17: write exact bounds into $\mathcal{C}$ for $\langle S, \mathrm{id}(E), \mathrm{LB/UB}\rangle$
18: **return** $(\mathrm{LB}(E, S), \mathrm{UB}(E, S))$

---

**Cost per call.** Let $a_S$ be the #affine leaves of the syntax DAG of $E$ reachable on $S$ (after e-graph sharing).

$$\text{LPs per call of EvalBoundsOnLeaf} \ \le\ 2\,a_S\cdot(1-h_{\text{LP}}), \qquad \text{SOCPs} \ \le\ 2\,a_S^{(\ell_2)}\cdot(1-h_{\text{LP}}). \tag{1}$$

All other constructors are $O(1)$ arithmetic on scalars.

### G.3 ENSURE_SIGN: Pseudocode and Complexity

---
**Algorithm 7** ENSURE_SIGN$(z, S)$ (local)

---
**Require:** scalar pre-activation $z$, leaf $S$
1: $(\ell, u) \leftarrow$ EvalBoundsOnLeaf$(S, z)$
2: **if** $u \le 0$ **then**
3:      commit negative branch on $S$; **return**
4: **else if** $\ell \ge 0$ **then**
5:      commit positive branch on $S$; **return**
6: **else**
7:      register (if absent) face $\{z \ge 0\}$ in $\mathcal{H}$
8:      split $S$ into $S^+, S^-$ and push to $\mathcal{L}$; **return**

---

**Per-invocation cost and bound.** One call makes at most one *threshold insertion* and one *split*. Bound cost $\le 2\,a_z(1 - h_{\text{LP}})$ LPs (from $z$); no other LPs are mandatory. Hence over the whole run,

$$\#\text{thresholds} \ \le\ B_{\text{guard}}, \qquad \#\text{splits} \ \le\ B_{\text{split}}.$$

### G.4 ENSURE_WINNER: Pseudocode, Pruning, and Complexity

---
**Algorithm 8** ENSURE_WINNER$(\{E_i\}_{i=1}^k, S)$ (local)

---
**Require:** candidates $\{E_i\}_{i=1}^k$, leaf $S$
1: **for** $i = 1$ to $k$ **do**
2:      $(\ell_i, u_i) \leftarrow$ EvalBoundsOnLeaf$(S, E_i)$
3: prune any $i$ with $u_i \le \max_j \ell_j$                    ▷ dominance, safe
4: **if** there exists a unique $i^\star$ with $\ell_{i^\star} \ge \max_{j \ne i^\star} u_j$ **then**
5:      commit candidate $i^\star$ on $S$; **return**
6: **else**
7:      choose $(p, q)$ maximizing $u_p - u_q - (\ell_p - \ell_q)$
8:      register (if absent) comparator $\{E_p \ge E_q\}$ and split $S$ into $S_{p\ge q}$ and $S_{q\ge p}$; **return**

---

**Per-invocation LP bound.** Let $k'$ be the remaining candidates after pruning. Then

$$\text{LPs} \ \le\ 2\Big(\sum_{i=1}^{k'} a_S(E_i)\Big)\cdot(1 - h_{\text{LP}}),$$

where $a_S(E_i)$ is #affine atoms under $E_i$ reachable on $S$. Each call inserts *at most one* comparator face and makes *one split*.

**Total comparator bound.** Because each call inserts a single face, total introduced comparators is $\le B_{\text{guard}}$; this *strictly* improves over static all-pairs $\sum \binom{k_v}{2}$.

## G.5   ENSURE_COMMON_REFINE: PSEUDOCODE AND COMPLEXITY

---

**Algorithm 9** ENSURE_COMMON_REFINE$(S; S_1, S_2)$

---

**Require:** current leaf $S$, guard sets $S_1, S_2$
1: $D \leftarrow (S_1 \cup S_2) \setminus S$
2: **while** $D \neq \varnothing$ and target sub-expressions not comparable/addable on $S$ **do**
3:     pick $\ell \in D$ by a policy (max-volume split; mid-face; etc.)
4:     register $\ell$ if absent and split $S$ accordingly
5:     update $D \leftarrow D \setminus \{\ell\}$

---

**Complexity.**   The loop performs at most $|S_1 \cup S_2| - |S|$ iterations; each iteration makes one split and may insert one new face. No mandatory LPs are required unless feasibility of a child is queried.

## G.6   B&B DRIVER: MICRO-POLICY, CERTIFICATES, AND COMPLEXITY

---

**Algorithm 10** JIT-B&B$(A, D, g)$: any-time solver on objective $g$

---

**Require:** SWT/JIT $A$, domain $D$ as $S_0$, scalar CPWL objective $g$
1: $\mathcal{L} \leftarrow \{S_0\}$; CERT$\leftarrow \varnothing$
2: **while** $\mathcal{L} \neq \varnothing$ and budgets not exhausted **do**
3:     $S \leftarrow \arg\max_{T \in \mathcal{L}} \text{UB}(g, T) - \text{LB}(g, T)$                           ▷ max-gap policy
4:     **if** $\text{LB}(g, S) \geq 0$ **then**
5:         mark $S$ as SAFE; CERT$\leftarrow$CERT$\cup\{(S, \text{LB})\}$; $\mathcal{L} \leftarrow \mathcal{L} \setminus \{S\}$; **continue**
6:     **if** $\text{UB}(g, S) < 0$ **then**
7:         solve $\min_{x \in C(S) \cap D} g(x)$ and output COUNTEREXAMPLE $(S, x^\star)$
8:     try ENSURE_WINNER/ENSURE_SIGN on $S$; else ENSURE_COMMON_REFINE
9:     push children to $\mathcal{L}$
10: **if** $\mathcal{L} = \varnothing$ **then**
11:     **return** PROOF with CERT
12: **else**
13:     **return** UNKNOWN with tightest-gap leaf $S^\star$ and its bounds

---

**Per-iteration LP/SOCP bound.**   Let $A_g(S)$ be the #affine atoms of $g$ reachable on $S$. Then one B&B iteration makes at most

$$2\, A_g(S) \cdot (1 - h_{\text{LP}}) + \Theta(1) \quad \text{LPs} \qquad (\text{plus SOCPs if } D \text{ involves } \ell_2 \text{ balls}),$$

for bound updates, plus the cost of the chosen refiner as in Sections G.3–G.5. If $\text{UB}(g, S) < 0$, an additional LP/SOCP is solved to obtain $x^\star$.

**Certificates (format).**   PROOF: a finite cover $\{(S_i, \text{LB}(g, S_i) \geq 0)\}$; COUNTEREXAMPLE: $(S, x^\star, g(x^\star))$; UNKNOWN: $(S^\star, \text{LB}, \text{UB})$.

## G.7   END-TO-END BOUNDS, AMORTIZATION WITH CACHING, AND PROOFS

**Theorem G.1** (Global guard/leaf/cardinality bounds). *With budgets* $(B_{\textit{split}}, B_{\textit{guard}})$ *and initial guard size* $|\mathcal{H}_0|$,

$$\#\text{leaves} \leq 1 + B_{\textit{split}}, \qquad |\mathcal{H}| \leq |\mathcal{H}_0| + B_{\textit{guard}}, \qquad \#\text{nodes/edges} \leq O(|Q_0| + B_{\textit{guard}}).$$

*Proof.* Each split replaces one leaf by two (leaf count $+1$). Only registering a *new* face increases $|\mathcal{H}|$. The shared SWT DAG grows only along edges touched by new faces; each face triggers $O(1)$ additions, giving $O(|Q_0| + B_{\textit{guard}})$. $\qquad\square$

**Theorem G.2** (Total LP/SOCP calls: worst-case and amortized). *Let $A^\star = \max_{S \in \mathcal{L}_{\max}} A_g(S)$ be the peak affine-atom count of g on any visited leaf, and let $\kappa$ be a constant bounding the number of sub-expressions whose bounds are refreshed per iteration. Then*

$$\#\mathrm{LP} \;\leq\; O\big((B_{\textit{split}} + B_{\textit{guard}} + |\mathcal{L}|) \cdot \kappa \cdot A^\star\big) \quad \textit{(worst-case, no cache)},$$

*and, with hit rate $h_{\mathrm{LP}}$,*

$$\mathbb{E}[\#\mathrm{LP}] \;\leq\; (1 - h_{\mathrm{LP}}) \cdot O\big((B_{\textit{split}} + B_{\textit{guard}} + |\mathcal{L}|) \cdot \kappa \cdot A^\star\big).$$

*The same holds for SOCPs when $\ell_2$ balls are present (with $A^\star$ restricted to affines whose domains couple to $\ell_2$ constraints).*

*Proof.* Per iteration bound from Section G.6; multiply by the number of iterations (bounded by $B_{\textit{split}} + B_{\textit{guard}} + |\mathcal{L}|$ since each iteration either commits or inserts a single face and splits once). Amortized form follows by linearity of expectation with hit rate $h_{\mathrm{LP}}$. $\qquad\square$

**Theorem G.3** (Comparator/threshold insertions and candidate complexity). *Let $C_{\max} = \max_S c(S)$ be the peak active-candidate count per leaf after dominance pruning. Under the single-face-per-step policy, the total number of winner comparisons introduced is $O(B_{\textit{guard}} + |\mathcal{L}|C_{\max})$.*

*Proof.* Each call of `ENSURE_WINNER` inserts at most one comparator. Across at most $|\mathcal{L}|$ active leaves and $C_{\max}$ candidates per leaf, the number of *attempted* comparisons is bounded by $|\mathcal{L}|C_{\max}$; only a subset materializes into registered faces, upper bounded by $B_{\textit{guard}}$. $\qquad\square$

**Corollary G.4** (Memory). *With guard uniqueness and e-graph sharing, memory is $O(|Q_0| + B_{\textit{guard}} + |\mathcal{L}|)$; reclaiming unreachable leaves and dead sub-expressions (reference counting) keeps this bound tight during execution.*

### G.8 Correctness of Caching and Monotone Tightening

**Lemma G.5** (Cache correctness under splits). *Let a cached pair $(\mathrm{LB}(E, S), \mathrm{UB}(E, S))$ be exact on leaf S. After splitting S into children $S^\pm$, the cached pair remains a valid* outer *sandwich on each child: $\mathrm{LB}(E, S) \leq \inf_{x \in C(S^\pm)} E(x) \leq \sup_{x \in C(S^\pm)} E(x) \leq \mathrm{UB}(E, S)$; recomputing on $S^\pm$ tightens these bounds (monotone).*

*Proof.* $C(S^\pm) \subseteq C(S)$; thus $\inf_{C(S^\pm)} \geq \inf_{C(S)}$ and $\sup_{C(S^\pm)} \leq \sup_{C(S)}$, giving a valid (possibly loose) sandwich. Exact recomputation on $S^\pm$ cannot be looser than the parent bounds. $\qquad\square$

**Theorem G.6** (Global monotonicity with caching). *If cached bounds are only replaced by tighter values and leaves are only refined by splits or commits, then the global anytime envelopes satisfy $\underline{A} \uparrow$ and $\overline{A} \downarrow$ pointwise during the whole run.*

*Proof.* Immediate from Lemma G.5 and the order-preserving constructors of the LB/UB engine (App. G.2): local tightening propagates monotonically through the DAG and the leaf aggregation ($\sup / \inf$). $\qquad\square$

**Takeaway.** The micro-algorithms in §G.3–G.6 ensure each step (*i*) inserts at most one face and makes at most one split, (*ii*) triggers a constant number of bound-refresh calls, and (*iii*) benefits linearly from caching. Theorems G.1–G.6 provide verifiable upper bounds and establish correctness and anytime monotone tightening for the full JIT–SWT loop.

## H Extended Experimental Details and Additional Results

This appendix expands the empirical section with full protocols, hyper-parameters, and additional visualizations for the three case studies: (A) FFN on MNIST, (B) CNN on CIFAR-10, and (C) GNN on the Karate Club graph.

Table 3: FFN—Local Lipschitz and robustness summary.

| | |
|---|---|
| Avg dynamic compile error (9760 samples) | $1.13 \times 10^{-6}$ |
| Global Lipschitz upper bound $L_{\text{upper}}$ | 22.9303 |
| Avg local-L (Top 50) | 6.6012 |
| Avg local-L (Bottom 50) | 2.2332 |
| ASR (Top 50, $\epsilon = 0.25$) | 100.00% |
| ASR (Bottom 50, $\epsilon = 0.25$) | 14.00% |

## H.1 EXPERIMENT A: FFN—LOCAL LIPSCHITZ VS. ROBUSTNESS

**Goal.** Validate that the JIT-SWT compiler recovers the local affine piece at a queried input and use it to estimate the *per-sample* local Lipschitz constant; evaluate its correlation with single-step adversarial robustness (FGSM).

**Setup.**

- **Data.** MNIST with standard train/test splits and normalization. *Code:* `transforms.ToTensor() + Normalize((0.1307,),(0.3081,))`.
- **Model.** FFN $784 \rightarrow 128 \rightarrow 64 \rightarrow 10$ with ReLU activations. *Training:* Adam, lr$= 10^{-3}$, 5 epochs, CE loss.
- **Compiler.** `SWT_Compiler` compiles the per-sample local affine piece $(W, b)$; the local Lipschitz is $\|W\|_2$.
- **Adversary.** FGSM with $\epsilon = 0.25$ in input space; attack is evaluated on the top/bottom-$N$ samples ranked by local Lipschitz (with $N = 50$).

**Protocol.** For each correctly classified test sample $x$, compile a local affine piece $f(x') \equiv Wx' + b$ in a neighborhood of $x$. Record the per-sample max absolute forward discrepancy $|f(x) - A(x)|_{\infty}$ and the local Lipschitz $\|W\|_2$. Then attack the top-$N$ (High-L) and bottom-$N$ (Low-L) groups using FGSM; report attack success rate (ASR).

**Results.** The average dynamic compilation error across 9760 correctly classified samples is $1.13 \times 10^{-6}$ (Figure 4). The empirical distribution of per-sample local Lipschitz values concentrates between 2 and 7 (Figure 5). The High-L group has mean 6.6012 vs. 2.2332 for Low-L. Under FGSM $\epsilon = 0.25$, ASR is 100% for High-L and 14% for Low-L (Table 3), confirming a strong positive correlation between local Lipschitz and adversarial susceptibility.

## H.2 EXPERIMENT B: CNN—TRANSLATION EQUIVARIANCE

**Goal.** Validate compiler applicability to CNNs and quantify translation equivariance under small integer shifts; visualize boundary effects.

**Setup.**

- **Data.** CIFAR-10 subset: 1000 train and 200 test images per class (normalized to mean/std 0.5).
- **Model.** `CNN_A`: Conv($3 \rightarrow 16, 3 \times 3$,s=1,p=1) $\rightarrow$ReLU$\rightarrow$ Conv($16 \rightarrow 32, 3 \times 3$,s=1,p=1) $\rightarrow$ReLU$\rightarrow$GlobalAvgPool$\rightarrow$Linear($32 \rightarrow 10$); trained for 5 epochs with Adam (lr$= 10^{-3}$), CE loss.
- **Shifts.** All $(\Delta x, \Delta y) \in \{-1, 0, 1\}^2 \setminus \{(0, 0)\}$ with zero padding upon rolling.
- **Test.** For 100 random test images, we compile a local piece before and after shift and compare class predictions; we report pass rate and a heatmap of failures over $(\Delta x, \Delta y)$.

**Results.** Mean per-sample dynamic compilation error over 100 images is $1.36 \times 10^{-7}$ (Figure 6). The equivariance pass rate over all shifts is **0.850**. Failures concentrate on diagonal shifts, consistent with boundary effects introduced by padding/stride (Figure 7).

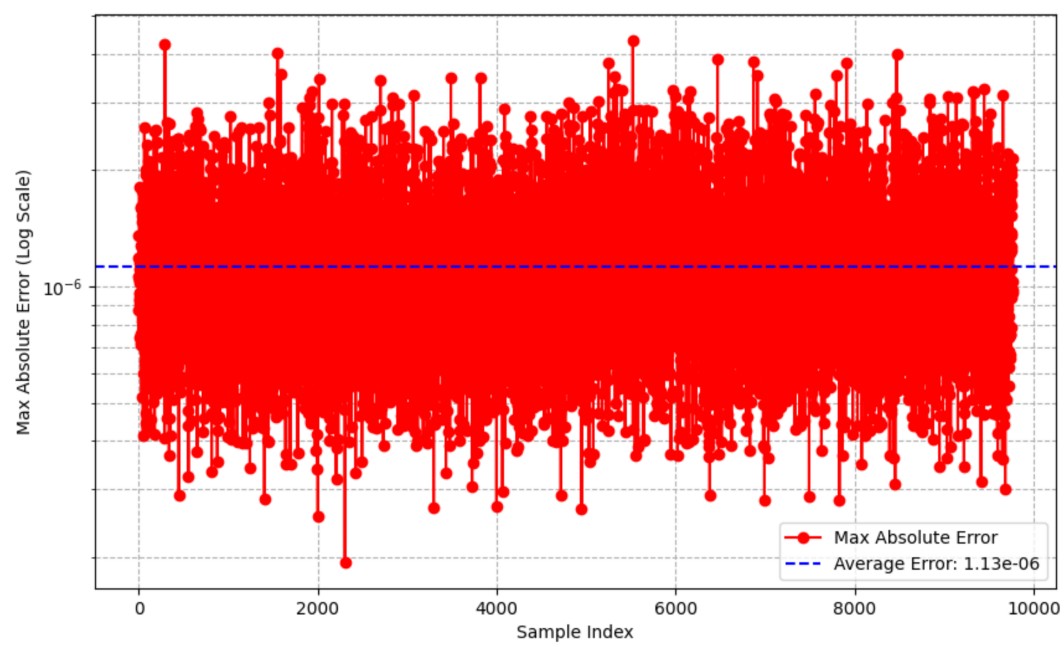

Figure 4: **FFN per-sample dynamic compilation error on MNIST** (log scale): max absolute discrepancy between compiled affine piece and PyTorch forward across correctly classified test samples; mean shown as dashed line.

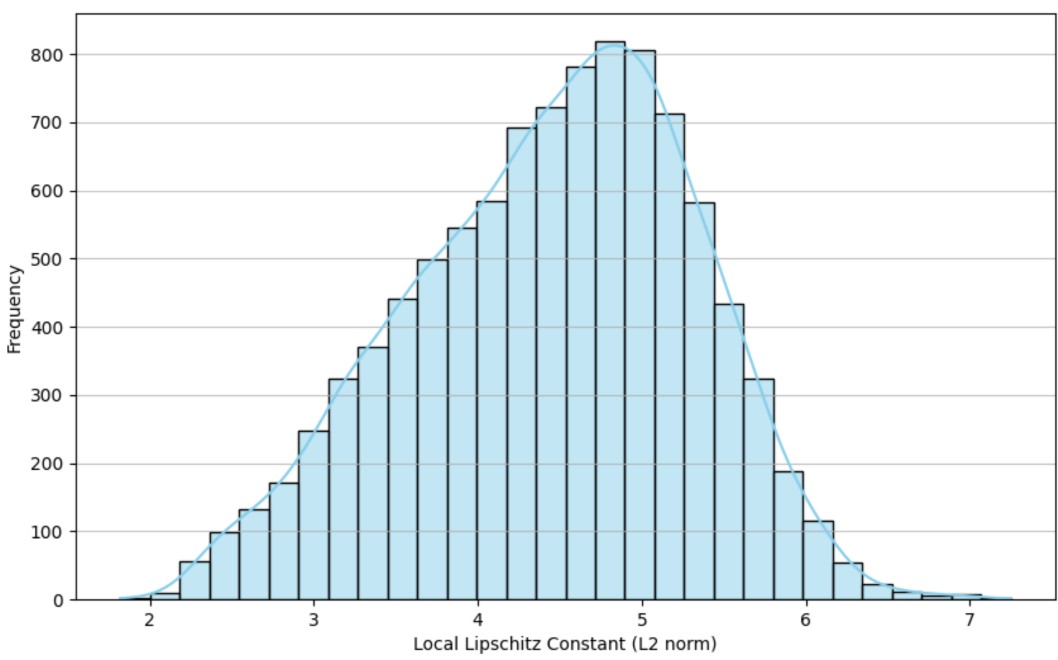

Figure 5: **Distribution of per-sample local Lipschitz constants** ($\ell_2$) on the MNIST test set.

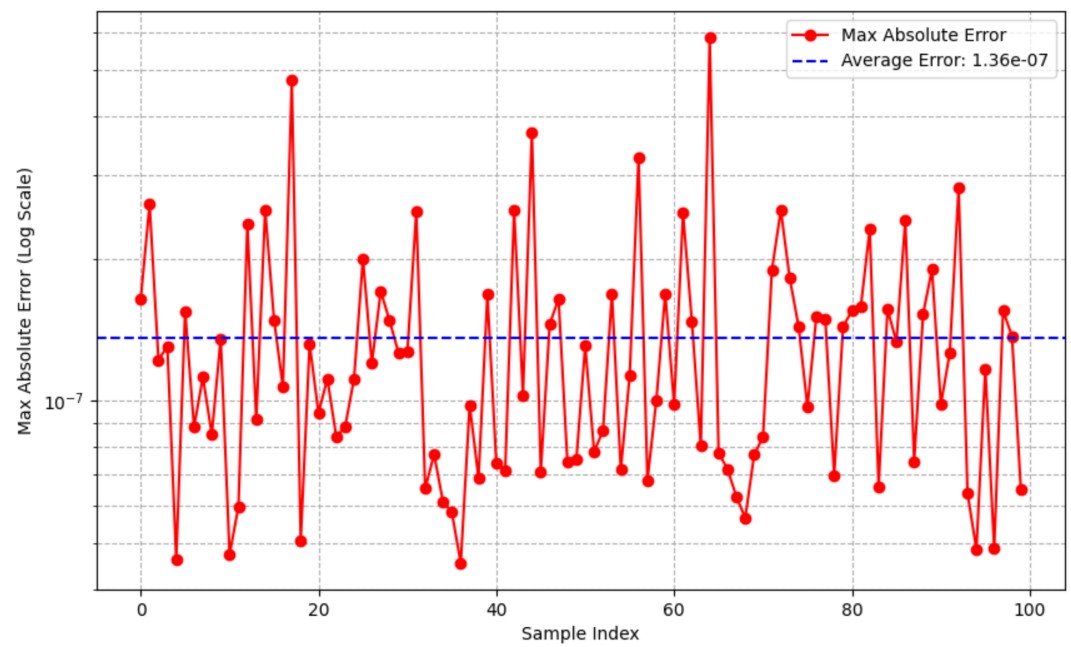

Figure 6: **CNN per-sample dynamic compilation error on CIFAR-10** (log scale).

### H.3 EXPERIMENT C: GNN—PROPERTY VERIFICATION AND CAUSAL INFLUENCE

**Goal.** (1) Verify compilation equivalence and permutation equivariance for a two-layer GCN; (2) quantify *maximum causal influence* (Imax) of hidden channels via intervention under bounded input perturbations; (3) confirm Imax as an importance signal by ablation.

**Setup.**

- **Graph.** Zachary's Karate Club ($|V|$=34). Features: node degree, clustering coefficient, and first 4 Laplacian eigenvectors (standardized).

- **Model.** GCNConv–ReLU–GCNConv–ReLU–Linear with hidden dim 16; trained 200 epochs (Adam, lr= $10^{-2}$) on the full graph.

- **Compilation & Verifications.** `SWT_Compiler` with node-wise shapes and normalized adjacency; (i) *Compilation equivalence:* random noisy inputs around the training features; (ii) *Permutation equivariance:* 50 random node permutations with degree-normalized adjacency recomputed per permutation.

- **Imax (B&B).** For each hidden channel $k \in \{0, \ldots, 15\}$, we intervene by zeroing channel $k$ at layer 2 and compute $I_{\max} = \sup_{\|X - X_0\|_\infty \le 0.1} \|F(X) - F_C(X)\|_\infty$ by a branch&bound solver over an $\ell_\infty$ ball (§H.3.1); we then ablate Top-5 / Bottom-5 channels and compare accuracies.

**Results.** Compilation equivalence (100 probes): mean error $2.17 \times 10^{-6}$, max $3.77 \times 10^{-6}$ (Figure 8). Permutation equivariance (50 perms): mean error $6.80 \times 10^{-15}$, max $1.07 \times 10^{-14}$ (Figure 9). The Imax B&B convergence for channel 0 is shown in Figure 10. Ranking all channels by Imax produces the Top-5 $\{0, 1, 8, 11, 12\}$ and Bottom-5 $\{4, 6, 7, 10, 13\}$ (Figure 11). Ablating Top-5 reduces accuracy from $100\%$ to $97.06\%$ (drop $2.94\%$), while ablating Bottom-5 has no impact ($100\%$ vs. $100\%$); see Figure 12 and Table 4.

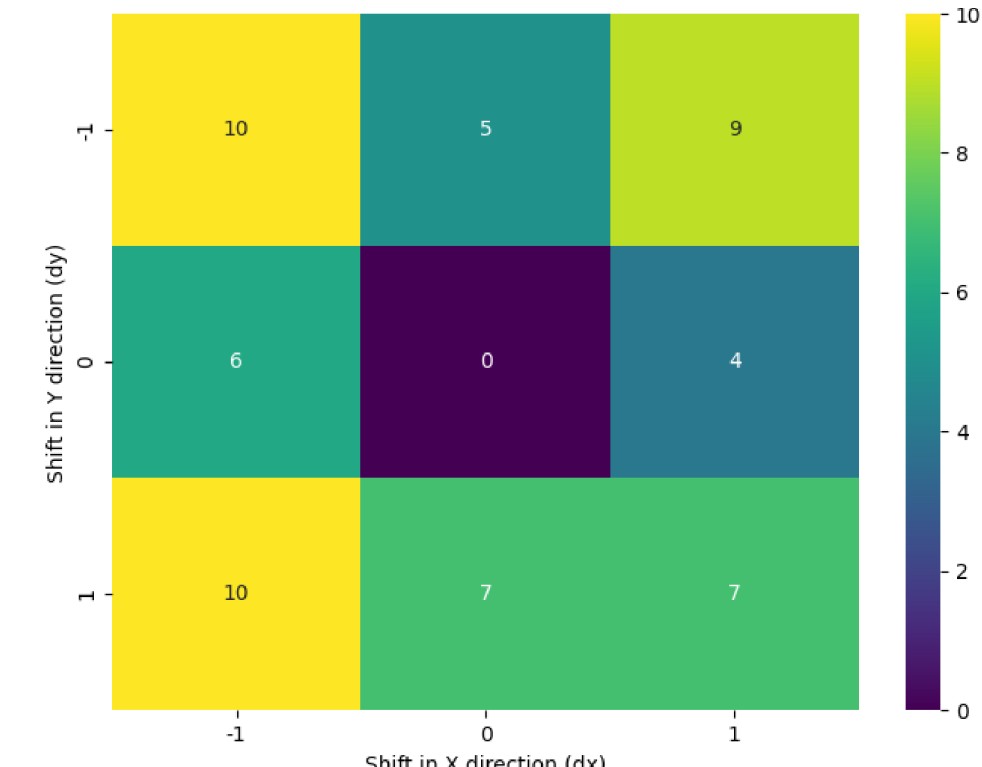

Figure 7: **Equivariance failure heatmap** over $(\Delta x, \Delta y) \in \{-1, 0, 1\}^2 - \{(0,0)\}$; brighter indicates more failures. Diagonals (e.g., $(-1,-1)$, $(-1,1)$) account for most failures due to boundary padding.

Table 4: GNN—verification and Imax ablation summary.

| | |
|---|---:|
| Max compilation error | $3.77 \times 10^{-6}$ |
| Mean compilation error | $2.17 \times 10^{-6}$ |
| Max equivariance error | $1.07 \times 10^{-14}$ |
| Mean equivariance error | $6.80 \times 10^{-15}$ |
| Baseline accuracy | $100.00\%$ |
| Accuracy after removing Top-5 | $97.06\%$ |
| Performance drop (Top-5) | $2.94\%$ |
| Accuracy after removing Bottom-5 | $100.00\%$ |
| Performance drop (Bottom-5) | $0.00\%$ |

### H.3.1 IMAX BRANCH&BOUND DETAILS

We consider the $\ell_\infty$ ball $\{X : \|X - X_0\|_\infty \leq 0.1\}$; its H-representation is $A = \begin{bmatrix} I \\ -I \end{bmatrix}$, $d = \begin{bmatrix} X_0 + \epsilon \\ -(X_0 - \epsilon) \end{bmatrix}$, with $\epsilon = 0.1$ and $X_0$ the baseline features (vectorized). On each B&B node, a feasible center is obtained via LP; both the original and intervened models are compiled at that point to local affine forms, whose difference yields per-facet lower/upper bounds on $\|F - F_C\|_\infty$. Subdomains are split along the dimension with largest interval range, halving at the mid-point, until budget/precision criteria are met. *Implementation:* `BranchAndBoundSolver` using SciPy HiGHS.

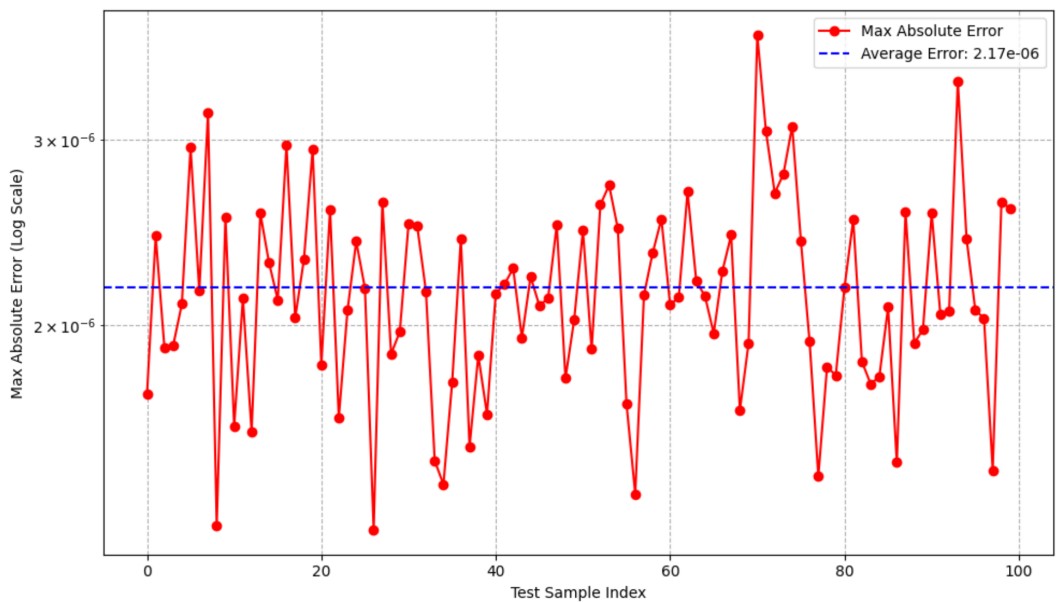

Figure 8: **GNN compilation equivalence**: per-probe max absolute error (log scale).

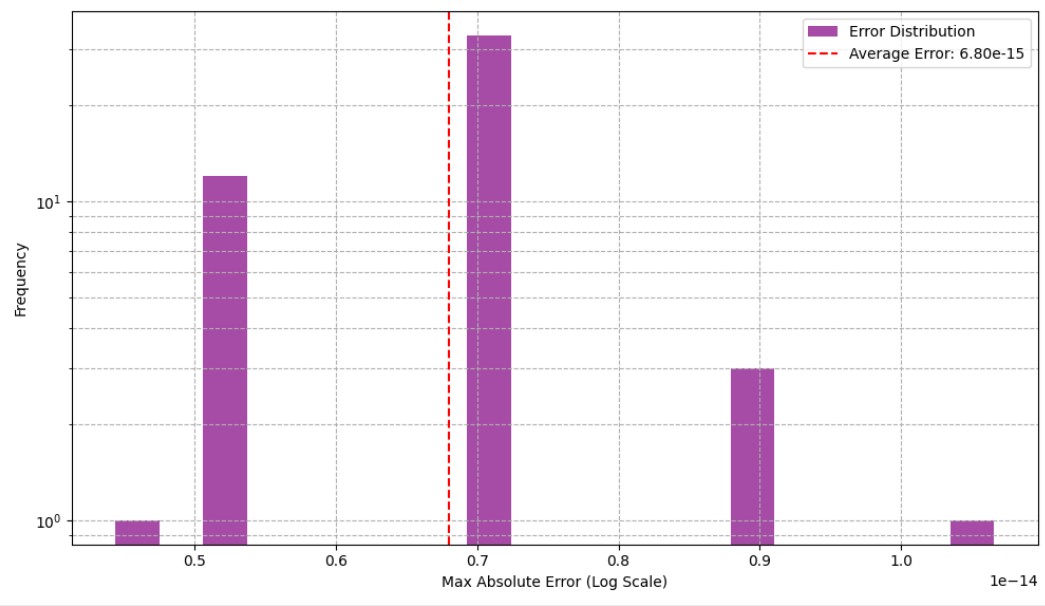

Figure 9: **Permutation equivariance** across 50 random permutations: histogram of max absolute errors (log scale).

# I  IMPLEMENTATION AND ROBUSTNESS DETAILS (REPRODUCIBILITY)

This section records engineering decisions that make our results easy to reproduce and stress-test. Wherever possible we point to the precise implementation sites in the released code.

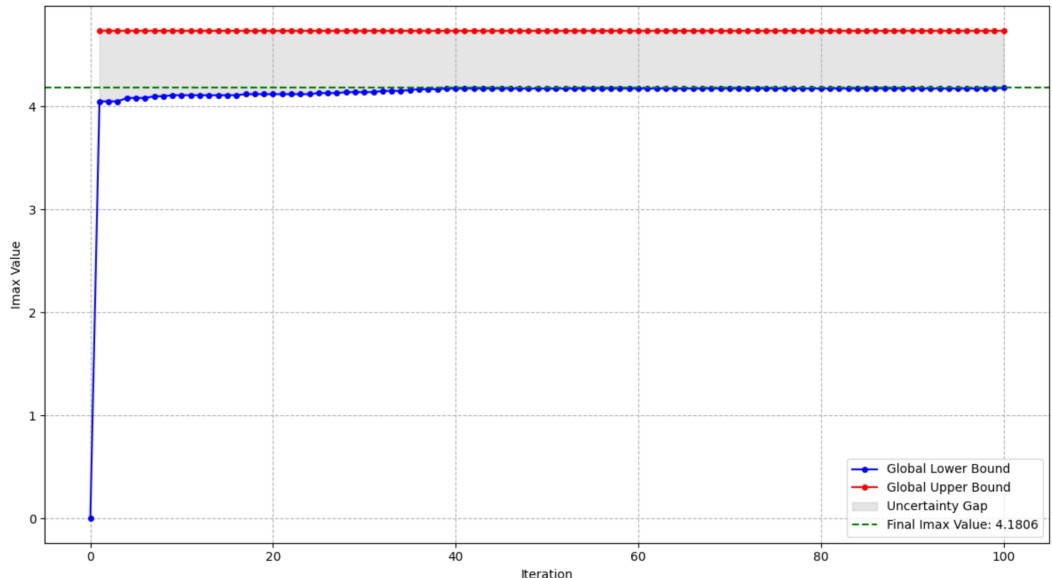

Figure 10: **Imax B&B convergence** for channel 0: global lower/upper bounds over iterations and uncertainty gap.

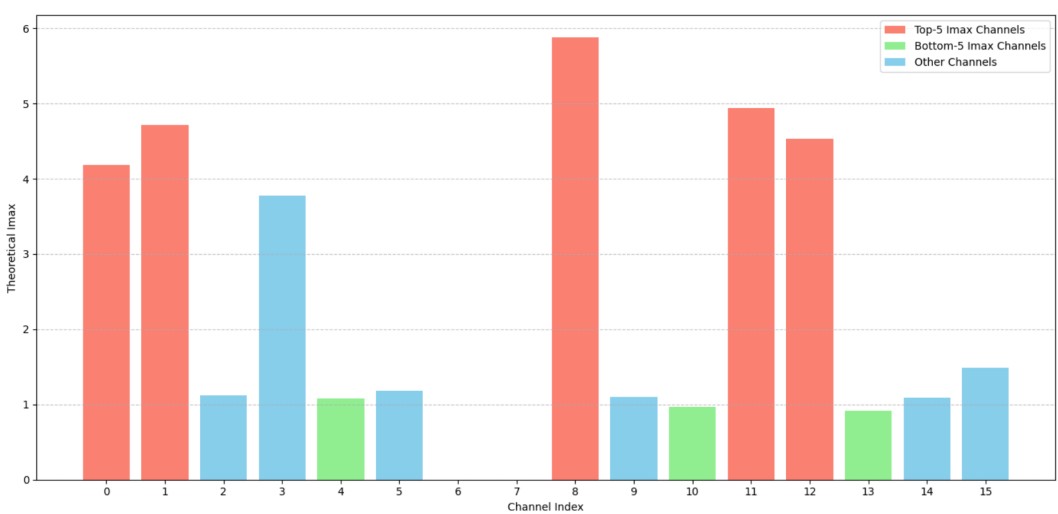

Figure 11: **Imax distribution** across 16 hidden channels; Top-5 (salmon) and Bottom-5 (light green) highlighted.

## I.1   RELU GUARD POLARITY AND TIE HANDLING

**Guard polarity.** When a pre-activation $z_i(x) = w_i^\top x + b_i$ passes through RELU, the active (non-negative) branch should add the guard

$$z_i(x) \geq 0 \iff (-w_i)^\top x \leq b_i,$$

and the inactive (non-positive) branch should add

$$z_i(x) \leq 0 \iff w_i^\top x \leq -b_i.$$

In the reference implementation, the polarity is encoded in `ReLuTransition.apply` by appending one linear inequality per unit to the current guard (H-polytope). Concretely, the active

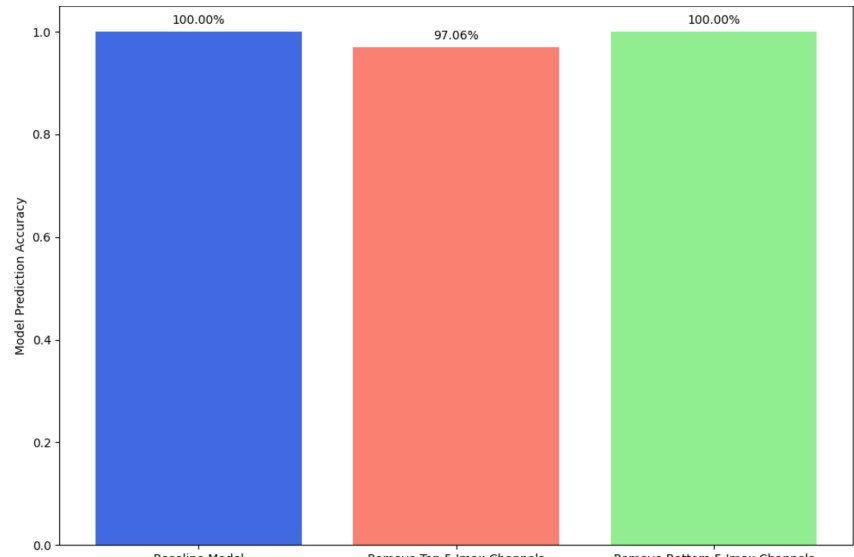

Figure 12: **Ablation study**: baseline vs. removing Top-5 vs. removing Bottom-5 channels.

branch correctly appends `A += -w_i, d += -b_i`, whereas the inactive branch should append `A += +w_i, d += -b_i`. If you see `d += b_i` on the inactive branch, flip the sign to `-b_i` to match the formulas above (one-line fix).

**Ties.** The theoretical semantics use *closed* guards on both sides of the hinge so that the $z_i(x) = 0$ hyperplane is covered from either side. In code we detect activity by a strict test on the numeric pre-activation (`pre_act_flat > 0`) and thus fall to the "negative" side at exact tie. If desired, you can (i) add a tolerance band and duplicate both guards when $|z_i(x)| \leq \tau$, or (ii) keep closed guards by switching to `>=` on the active side and `<=` on the inactive side; the global tolerance constant lives at `FP_TOLERANCE = 1e-5`.

**Where to patch.** `compiler.py` → `ReLuTransition.apply` and the file-level constant `FP_TOLERANCE`.

## I.2 CONSISTENT SCIPY.LINPROG BOUNDS

We solve small LPs inside the branch-and-bound (B&B) loop. **Recommendation:** pass explicit per-variable bounds as $\text{bounds} = [(\text{None}, \text{None})] \times n$ for all calls. This avoids backend-specific behavior and makes the code portable across SciPy versions.

*Where used.* (i) In `Guard.is_feasible()`, bounds are already constructed per dimension. (ii) In `BranchAndBoundSolver._get_bounds_in_guard()` we set `bounds=[(None,None)] * input_dim` (good). (iii) In `BranchAndBoundSolver.solve()` the range queries use `bounds=(None,None)`; although accepted by SciPy as a global bound pair, we recommend unifying it to the list form for consistency. All three sites are in `compiler.py`.

## I.3 CONVOLUTIONAL JACOBIANS: PRACTICALITY AND A MATRIX-FREE OPTION

**What the release does.** The current `Conv2dTransition.apply` materializes the linear map of a convolution by building an *im2col-style* matrix $W_{\text{conv}} \in \mathbb{R}^{(C_{\text{out}} H_{\text{out}} W_{\text{out}}) \times (C_{\text{in}} H_{\text{in}} W_{\text{in}})}$ and composing it with the running affine piece $(W, b)$. This is exact and works well at the CIFAR-10 scale we use $(32 \times 32)$ but can be memory-heavy for larger inputs.

**Matrix-free alternative (recommended for large inputs).** Replace the explicit $W_{\text{conv}}$ with a *linear operator* that supports the actions $v \mapsto W_{\text{conv}} v$ and $u \mapsto W_{\text{conv}}^\top u$ using PyTorch's conv primitives:

$$W_{\text{conv}} v \equiv \text{conv}(v \text{ reshaped}), \qquad W_{\text{conv}}^\top u \equiv \text{conv\_transpose}(u \text{ reshaped}).$$

Then propagate the affine piece without ever constructing $W_{\mathrm{conv}}$ explicitly. This drop-in operator fits our automaton because every composition step only needs the product $W_{\mathrm{new}} \cdot W$ and $W_{\mathrm{new}} \cdot b$. In practice, implement a small wrapper class with `matvec` and `rmatvec`, and gate between the dense and matrix-free paths by input size. The convolution path in the CNN experiment invokes `SWT_Compiler(..., initial_shape=(3,32,32))` and then hits `Conv2dTransition` during `trace`.

*Note.* Our GCN path already avoids materializing the graph conv Jacobian symbolically; it computes the linear map by probing the layer with basis vectors once and reuses the result, which is tractable on Karate Club. For larger graphs, the same matrix-free pattern applies.

## I.4 CNN TRANSLATION FILL VALUE AFTER NORMALIZATION

When CIFAR-10 images are normalized to mean 0.5 and std 0.5, the dynamic range is roughly $[-1, 1]$. In the `shift_image` routine we currently zero-fill the wrapped border after `torch.roll`, which injects a mid-gray value and slightly biases the equivariance test at the padded faces. To emulate "black" padding in the normalized space, set the border to $-1$ instead of 0:

```python
def shift_image(x, dx, dy):
    x_s = torch.roll(x, shifts=(dy, dx), dims=(-2, -1))
    fill = -1.0  # black in normalized [-1, 1]
    if dy > 0:   x_s[..., :dy, :] = fill
    elif dy < 0: x_s[..., dy:, :] = fill
    if dx > 0:   x_s[..., :, :dx] = fill
    elif dx < 0: x_s[..., :, dx:] = fill
    return x_s
```

This function lives in `exp2.py` under the CNN experiment.

## I.5 ENTRY POINT, SEEDS, AND DETERMINISTIC SETTINGS

**Single entry point.** All experiments can be reproduced via

$$\text{python run.py --exp } \{1, 2, 3\}$$

for FFN/CNN/GNN respectively. The dispatcher is defined in `run.py`.

**Seeding and determinism.** We fix a global seed (`SEED = 2025`) and set deterministic flags for CUDA/cuDNN at module import time; this covers PyTorch RNGs and reduces non-determinism during convolutions. See the seed block at the top of `compiler.py`.

**Data loaders.** The FFN/CNN training loops use standard PyTorch data loaders; in the CIFAR-10 setup we set `num_workers=2`. For byte-for-byte determinism, set `num_workers=0` and `shuffle=False` (or fix the sampler seed) during training; evaluation and all compiler-based analyses are deterministic given the saved weights. The MNIST/CIFAR-10 pipelines and model definitions reside in `exp1.py` and `exp2.py`; the GNN pipeline (Karate) is in `exp3.py`.

**B&B solver knobs.** The Imax computation uses a compact branch-and-bound with LP relaxations; `max_iter` and `tolerance` control the budget/precision trade-off. See `BranchAndBoundSolver` and its use sites in the GNN experiment.

