# OpenReview forum: "Just-in-Time Piecewise-Linear Semantics for ReLU-type Networks"
_ICLR.cc/2026/Conference — Submitted to ICLR 2026_

### Official Review · Reviewer_m6BL · 2025-10-23

**Soundness:** 3
**Presentation:** 1
**Contribution:** 3
**Rating:** 4
**Confidence:** 2

**Summary:**

The paper introduces a new neural network reasoning method based on a compilation of the network into an automaton with polyhedral edge guards and node and edge weight matrices. The authors show that any neural network with piecewise linear activation functions can be represented exactly in this automata language. A dynamic ("just in time") expansion of the automaton is developed to analyze different properties (e.g., robustness, equivalence, etc) over the network's input-output behavior. To this end, the method features mechanisms to propagate input constraints and to introduce dynamically additional hyperplanes refining the guards if necessary. A thorough theoretical analysis studies a wide range of properties of this method, such as correctness and completeness for different analysis tasks, and the connection to concepts from geometry. A small empirical study show cases the flexibility of the developed method.

**Strengths:**

Owed to requirements in many applications, the formal analysis of learned functions, particularly neural networks, is receiving and has received significant attention in ML research in the recent years. This paper introduces a (to my knowledge) completely novel idea in this context. The algorithm is highly flexible in tackling a wide range of different analysis tasks. The paper evaluates this new method thoroughly from a theoretical perspective. Detailed proofs are provided in an appendix.

**Weaknesses:**

The write-up is very dense. Essentially, the main text is just a lose connection of different theoretical results, without providing a clear coherent story, or even explaining how the proposed method works exactly. Some of the introduced processing steps seem to be needed for some specific analysis task only, yet the paper lacks any sort of explanation when some steps are executed, in which order, and what context they are relevant. An overview of the overall method such as in the form of pseudocode would greatly improve comprehensibility. As it stands now, the paper is impossible to digest for non-expert readers. The general telegraph style of the write-up is not exactly beneficial for the clarity either.

The purpose of the experimental evaluation is also unclear. Given that the authors propose a new method for existing and well-researched problems, I would expect to see a comparison to state-of-the-art methods. Nevertheless, the authors just ran their methods on three different analysis tasks and different network architectures. This might show case the versatility of the proposed method, but apart from this it is not clear what to conclude from the results. Questions like the following remain open, but are really what should be shown in the empirical study: How does the method scale? Are the shown results in any sense "impressive", e.g., in terms of dealing with network sizes so far beyond reach or handling properties that were so far not considered? How is the method placed with respect to other approaches? How effective is the algorithm, in terms of bound propagation and in terms of the number on-demand splits generated vs. unfolding the complete computation tree.

The paper certainly makes a major contribution, but given its current presentation and the unclear experiment design, it needs another thorough revision.

**Questions:**

1. Have you compared your algorithm to other state-of-the-art methods, and if so, how does it perform?

---

> ### Author Response · Authors · 2025-12-03
>
> We thank the reviewer for identifying key issues regarding the narrative clarity, lack of an algorithmic overview, and the specific purpose of the experiments. We fully agree with these concerns regarding readability and evaluation presentation, and we have made targeted revisions in the updated manuscript:
>
> **1. Narrative Structure and Overview**
> In the Introduction, we have completed the main framework description (Figure 1: Network → SWT (static semantics) → JIT-SWT (on-demand refinement) → Unified Geometry/Verification/Causal Tasks). We also added Table 1 to index core conclusions and theorem labels, preventing the main text from becoming a scattered list of theorems.
>
> **2. Algorithm Execution and Steps**
> Regarding "how the method works exactly," the revised manuscript provides the end-to-end Branch-and-Bound driver pseudocode (Algorithm 1) in §3 to clarify the execution order.
>
> * **Process:** In each round, the algorithm selects the GuardSet with the maximum gap. It first checks Lower/Upper Bounds (LB/UB), which may directly yield certificates or counterexamples.
> * **Refinement:** If needed, it invokes three atomic refiners: `ENSURE_SIGN` (for undecided gate signs), `ENSURE_WINNER` (for Max/MaxPool candidates), and `ENSURE_COMMON_REFINE` (for scenarios requiring expression alignment, such as equivalence checks).
> * **Guarantees:** We prove anytime soundness, exactness upon local full refinement, and the "no regression" property (DYN-1..3), alongside budgeted complexity upper bounds and the correctness of dominance pruning (DYN-4/5).
>
> **3. Experimental Purpose and SOTA Comparison**
> We clarify that the experiments are designed to demonstrate **sanity + versatility**:
> * **Sanity:** We first verify that JIT refinement recovers local affine fragments consistent with the original model to machine precision (FFN/CNN/GNN forward errors $\approx 10^{-6}-10^{-7}$).
> * **Versatility:** We use the *same carrier* to trigger three distinct analyses: local Lipschitz-FGSM correlation, CNN translation equivariance failure localization (at padding boundaries), and GNN permutation equivariance with Imax-guided ablation. This confirms that the "unified carrier + on-demand geometric refinement" is indeed reusable across tasks.
> * **Comparison:** The current manuscript does not include a systematic head-to-head runtime comparison against mainstream verifiers (like $\alpha$-CROWN), and we accept this suggestion. However, we emphasize that existing SOTA methods often perform highly engineered optimizations for a *single task* (especially robustness). Our core contribution is providing a CPWL semantic carrier and an on-demand refinement mechanism reusable across robustness, equivalence, equivariance, and causal intervention tasks. Therefore, comparisons require designing "comparable metrics" (e.g., normalized splits/LP call counts) rather than just wall-clock time on a single property.

---

### Official Review · Reviewer_4K8x · 2025-10-27

**Soundness:** 3
**Presentation:** 2
**Contribution:** 3
**Rating:** 6
**Confidence:** 3

**Summary:**

The paper introduces Just-in-Time Symbolic Weighted Transducers (JIT-SWT) as an execution semantics for CPWL neural networks.

Rather than completely expanding a ReLU network's exponentially large region decomposition, JIT-SWIT builds a guarded CPWL transducer and refines it on demand during the analysis.

The framework support sound any time envelopes, exactness on fully refined cells, formal reasoning tasks, and some experiments on FFNs, CNNs, GNNs.

**Strengths:**

Originality: The formalization seem novel and the idea of bridging algebraic formalisms and neural verification seems new.

Quality: Comprehensive theoretical framework with proofs of soundness, decidability, and budgeted complexity bounds.

**Weaknesses:**

Accessibility: The main text is quite dense and does not provide enough background, motivation, or illustrations for several of the definitions and proofs in the paper. A brief paragraph before each definition or at the beginning of each subsection providing an intuition on why the concepts are needed, or how they work at an intuitive level would be appreciated.

Scalability: Experiments are on small networks and restricted settings. A clearer discussion on when this can be applied and when it cannot would be appreciated.

Related work: It is not clear if this is the only solution out there for neural verification, geometry, and causal analysis. Are there other solutions that could be used for this? If yes, how does this compare to them? In which regards is this solution better?

**Questions:**

Is this approach the first one of its kind? Have other solutions been proposed in the past? How does JIT-SWT compare to them? If it is the first of its kind, what were the challenges in the past that prevented JIT-SWT from happening? What were the insights that allowed to overcome these challenges?

What are the scalability limits of this approach?

Can more concepts be moved into the appendix so there is room for more intuitions and examples that can help readers who are not very familiar with this topic understand the quality of this work?

---

> ### Author Response · Authors · 2025-12-03
>
> Thank you for the thoughtful review. We address the main questions below.
>
> **(1) Novelty / related work.** Prior neural verification methods (SMT/MILP/convex relaxations/branch-and-bound) are largely *solver-centric*: they answer one property end-to-end by repeatedly reasoning about constraints, but do not expose a reusable symbolic carrier that supports *multiple* downstream tasks (region extraction, Jacobians, Lipschitz, equivariance, interventions) with shared structure and incremental refinement. In contrast, JIT‑SWT is an *execution semantics*: it compiles a ReLU-type network into a guarded CPWL transducer with a shared guard library and a shared lazy expression DAG, and refines only where needed while maintaining sound anytime envelopes and exactness on fully refined leaves (DYN‑1/2) plus budgeted complexity bounds and dominance pruning (DYN‑4/5).
>
> **(2) Why this was hard before / key insight.** The main blocker for symbolic CPWL formalisms has been *expression/region explosion* and the need to pre-materialize huge comparator sets. Our key insight is to separate structure sharing (single SWT/e-graph) from geometry (a global, unique guard library) and to insert threshold/comparator faces only when a visited GuardSet cannot decide sign/winner/common-refinement. This yields monotone progress with certificates, without global enumeration.
>
> **(3) Scalability limits.** Worst-case complexity remains exponential in the number of reachable CPWL cells (an inherent limitation of exact CPWL reasoning). What JIT‑SWT guarantees is that *actual cost* is proportional to performed work: linear in #splits/#new faces and the resulting local LP/SOCP calls, and it can stop anytime with sound TRUE/FALSE/UNKNOWN outcomes.  Practically, JIT‑SWT is strongest when the queried property is “local” (e.g., robustness around a point, effective-domain equivariance, bounded interventions) and weakest when the objective forces near-global region coverage or very large conv Jacobians (for which we outline a matrix-free variant).
>
> **(4) Accessibility / restructuring.** We agree and moved additional proof-heavy and micro-algorithmic details to the appendix, freeing main-text space for (i) intuition paragraphs before key definitions/refiners and (ii) a running low-dimensional example that walks through ENSURE SIGN/WINNER and the envelope tightening behavior.

---

### Official Review · Reviewer_WzSZ · 2025-11-01

**Soundness:** 3
**Presentation:** 3
**Contribution:** 3
**Rating:** 6
**Confidence:** 4

**Summary:**

The paper proposes a "Just-in-Time" (JIT) Symbolic Weighted Transducer (SWT) framework to model ReLU-based neural networks. Unlike traditional static analysis that suffers from exponential explosion by eagerly materializing all regions, this approach performs on-demand refinement. It maintains sound anytime lower/upper envelopes and guarantees exactness when cells are locally fully refined. The authors demonstrate that this unified "carrier" can support diverse tasks—including region extraction, exact Jacobian computation, Lipschitz estimation, robustness verification, and causal influence analysis—by reducing them to sequence of local Linear Programming (LP) or Second-Order Cone Programming (SOCP) problems.

**Strengths:**

1. Unified Theoretical Framework: The primary strength is the unification of disparate geometric and formal tasks into a single symbolic carrier. Instead of specialized algorithms for Lipschitz estimation vs. robustness verification, the JIT-SWT provides a common substrate where these become differing objectives on the same refinement engine.
2. Anytime Guarantees & Monotonicity: The proofs for sound anytime envelopes (DYN-1) and monotonic progress without regression (DYN-3) are theoretically satisfying. This allows the method to act as both an exact verifier (given enough time) and an approximate certifier under budget constraints.
3. Budgeted Complexity: Explicitly bounding the complexity (number of active GuardSets, library size) linearly by the number of splits and inserted guards (DYN-4) is a strong theoretical contribution, distinguishing it from purely exponential worst-case static analysis.
4. Decidability: The proof that properties expressible with finite affine/SOC constraints are decidable under fair JIT refinement strategies (DYN-6) places the method on solid formal ground.

**Weaknesses:**

1. Scalability Concerns: While "JIT" mitigates immediate exponential explosion, the reliance on solving LPs/SOCPs in the inner loop of the Branch-and-Bound driver 8 represents a significant computational bottleneck. The experimental validation is limited to very small-scale problems (MNIST FFNs, subsets of CIFAR-10 with tiny CNNs, Karate Club GNNs). There is little evidence this approach can scale to standard Deep RL policy networks (e.g., ResNet-50 size or Transformer-based policies), where the number of active paths can be overwhelmingly large even for local properties.
2. Restriction to CPWL: The framework is rigorously defined only for CPWL networks (affine + Max/ReLU type gates). Modern state-of-the-art architectures, particularly in RL (Transformers, world models), rely heavily on non-CPWL operations like Softmax, Attention, and LayerNorm. The paper notes this limitation and suggests "sound PL relaxations" as future work11, but without this, its applicability to current high-impact models is severely limited.
3. Practicality of "Exact" Metrics: While exact Lipschitz or $I_{max}$ computation is theoretically appealing, in many DL/RL applications, tight upper bounds (obtainable faster via specialized abstract interpretation methods like $\alpha,\beta$-CROWN) are often sufficient. The paper does not extensively compare the cost-benefit ratio of JIT-SWT against state-of-the-art incomplete verifiers on larger benchmarks.

**Questions:**

1. You mention "certified PL lifts for multiplicative modules" (like Attention/Softmax) as future work. Do you envision these relaxations fitting neatly into the current SWT guard structure, or will they require a fundamental redesign of the semantic carrier to handle fundamentally non-linear/non-convex regions efficiently?
2. Comparison with Incomplete Verifiers: How does JIT-SWT compare in wall-clock time against state-of-the-art incomplete verifiers (such as $\alpha,\beta$-CROWN) when the goal is only to find a sound bound rather than an exact one? Does the overhead of the unified SWT structure make it competitive for purely bounded verification tasks on larger networks?
3. Sensitivity to Scheduler & Budget: The decidability guarantees rely on a "fair scheduler" and sufficient budgets. In practice, how sensitive is the quality of the returned bounds to the choice of micro-policy (e.g., max-gap vs. round-robin) when operating under tight computational budgets?

---

> ### Author Response · Authors · 2025-12-03
>
> We thank the reviewer for the affirmation and the critical inquiries. We clarify our positioning and respond point-by-point to the three main concerns and three questions:
>
> **1. Scalability and the LP/SOCP Bottleneck:**
> The core of JIT-SWT is not to "eagerly enumerate all linear regions before solving," but to strictly limit solving to local refinement on visited leaf GuardSets. Theoretically, we provide budgeted linear bounds: the number of active GuardSets, the size of the guard library, and the count of LP/SOCP calls all grow linearly with splits and added hyperplanes (DYN-4). Furthermore, dominance pruning guarantees the safe removal of candidates that cannot win within a leaf (DYN-5), significantly reducing the comparison burden of Max branches. Even when the budget is insufficient, the system returns monotonically tightening sound anytime lower/upper envelopes (DYN-1/3). On the implementation level, we also provide caching and a matrix-free path for convolutional Jacobians to avoid the explicit materialization of large matrices (Appendix I.3).
> We agree that pushing "exact decidability" to the scale of ResNet-50 or Transformers is not the empirical claim of this paper. Our contribution lies in providing a reusable unified semantic carrier and an on-demand refinement mechanism, where the cost is proportional to the size of the visited subdomains rather than exponentially exploding from the start.
>
> **2. CPWL Restrictions and Multiplicative Modules:**
> This paper intentionally scopes its focus to CPWL networks (affine + ReLU/Max type gates) to provide rigorous semantics, decidability, and certificates. For non-CPWL operators like Softmax, Attention, or LayerNorm, the "certified PL lift" we envision does not require overturning the SWT guard/leaf structure. The approach is to introduce certified linear or second-order cone outer approximations (e.g., McCormick or SOC bounds) into the leaf subproblems, allowing these modules to interface with existing oracles as "primitives with certified sound LB/UB." Accordingly, the guarantee of "final exactness/decidability" would naturally degrade to "certified sound bounds" for these components.
>
> **3. Comparison with Incomplete Verifiers (e.g., $\alpha$-CROWN):**
> JIT-SWT's LB/UB calculation does not mandate the use of exact LP/SOCP. The text explicitly permits arbitrary sound relaxations, which can be upgraded to exact solving when needed (the oracle is pluggable). Therefore, when the goal is merely to quickly obtain sound bounds, one can directly use bounds like $\alpha$-CROWN as the leaf oracle. Local refinement and exact solving are triggered only when "counterexamples or local exact geometry (Jacobians, exact Lipschitz, causal Imax, etc.)" are required. In other words, JIT-SWT is designed to house both "fast incomplete bounds" and "on-demand exactness" within the same reusable geometric carrier, rather than to replace specialized bounding tools.
>
> **4. Scheduler and Budget Sensitivity:**
> We emphasize two points: (i) Sound anytime properties and monotonicity do not depend on the micro-policy (DYN-1/3); different strategies only affect convergence speed. (ii) Completeness relies on "fair scheduling." We default to using the max-gap policy (Alg. 1) to directly minimize global uncertainty. Under tight budgets, this can be mixed with round-robin or volume-based heuristics to improve robustness and coverage balance.

---

### Official Review · Reviewer_9ZJr · 2025-11-03

**Soundness:** 2
**Presentation:** 1
**Contribution:** 3
**Rating:** 2
**Confidence:** 5

**Summary:**

The paper proposes JIT-SWT, a just-in-time semantics for analyzing ReLU-type networks as continuous piecewise-linear (CPWL) functions. The main idea is to compile networks into guarded transducers with shared guards, then refine on-demand rather than enumerating all fragments upfront. This method could avoid the computational explosion and will benefit the scalability aspect of tasks, including computing Lipschitz constant, robustness checking, etc. The main contributions are summarized as follows:
1. The framework for calculus and static compilation rules for CPWL NNs.
2. JIT refinement with soundness guarantee and complexity bounds.
3. Geometry and verification procedures leveraging the shared carrier
4. Experiments on MNIST/CIFAR-10/Karate Club demonstrating compiler correctness and correlations with network properties

**Strengths:**

- The problem formulation is clear. Unrolling ReLU layer-by-layer results in exponentially growing computational complexity. This paper proposed a principled approach with formal guarantees.
- Guard-sharing and unified representation are neat contributions. Inserting comparator faces only when needed yields better computational complexity. Having a single object support multiple verification tasks (Lipschitz, robustness) through a common LP/SOCP interface could meaningfully simplify verification pipelines.

**Weaknesses:**

- The paper is hard to read. The main text still feels like navigating a theorem catalog rather than a coherent story. The paper is dense with theorem families and heavy notations. It lacks descriptions of theoretical results and their role in the proposed method. Table 1 helps a bit, but a major streamlining pass is needed.
- Results AF-1 through AF-5 appear to be standard CPWL preliminaries (closure, continuity, differentiability). The paper should explicitly delineate which theoretical results are novel versus which are foundational background, making it easier to assess the core technical contributions.
- The experiments lack comparisons with existing works. We have no information about runtime and memory consumption relative to state-of-the-art tools, like benchmarks in VNN-COMP. Without baseline comparisons, the practical value remains unclear.

**Questions:**

1. Can the authors provide a clearer roadmap of how the theorem families (AF, SWT, DYN, GEO, VER, CAU) relate to each other and to the overall method? Specifically, which results are novel contributions versus foundational CPWL preliminaries?
2. The scope of the paper in the literature is lacking; there are literally hundreds of papers on neural network verification now, and the paper cites <10, none more recent than 2020. Either can the authors place the paper better in the existing vast literature and compare it qualitatively to what exists, or add comparisons against VNN-COMP benchmarks (and participating tools, eg, Marabou, alpha-beta-Crown, nnenum, NNV, NeuralSAT, Pyrat, etc.) and report wall-clock time, memory usage, and certified accuracy compared?
3. The decidability result (DYN-6) assumes finite refinement and fairness. In practice, how often does the system return UNKNOWN under realistic budgets?

---

> ### Author Response · Authors · 2025-12-03
>
> Thank you for the careful review. We address the main concerns below.
>
> #### Roadmap and what is novel
>
> We agree the theorem families need a clearer “story”. In the revised draft we added a contribution map (Fig.1) and a one-page theorem index (Table 1) that explicitly connects AF→SWT→DYN→(GEO/VER/CAU). Concretely:
>
> - AF-1..5 are CPWL preliminaries are closure/continuity/a.e. differentiability/composition, and are included only to make the semantics self-contained.
> - The novel technical core starts at SWT-1 are network→guarded transducer with shared guard library, pointwise equivalent.
> - DYN-1..6 are JIT refinement objects + three refiners + anytime sound envelopes + no-regression + budgeted linear bounds in #splits/#new guards + dominance pruning + decidability under finite refinement/fairness.
> - The task layer (GEO/VER/CAU) showing how a single shared carrier yields region/Jacobian extraction, exact/certified Lipschitz and LP/SOCP-based robustness/equivariance/equivalence and causal influence.
>
> #### Presentation / readability.
>
> We fully agree the original readability was not acceptable. We have performed a major narrative pass:
>
> - Moved most technical proofs into the appendix.
> - Added the “contribution map + theorem index” to guide readers.
> - Rewrote the introduction/method overview to explain *why each theorem family exists* (static semantics → dynamic refinement guarantees → downstream procedures), instead of presenting results as a catalog.
>
> #### Literature scope and missing baselines (VNN-COMP / tools).
>
> We acknowledge the current section's insufficient citations and recognize that baseline comparisons are crucial for assessing practical impact. Our goal is not to create “yet another end-to-end verifier,” but rather to build reusable symbolic carriers that enable *shared guard structures to support multi-task processing* through on-demand refinement (rather than re-solving each task from scratch). To this end, we have substantially expanded our literature citations beyond 2020 and conducted a qualitative positioning of solver-oriented approaches (SMT/MILP/branch-and-bound/convex relaxation). However, due to time constraints, we will not provide supplementary benchmark-style comparisons in this work.
>
> #### “UNKNOWN” frequency under budgets (DYN-6).
>
> DYN-6 employs a tri-valued design: UNKNOWN indicates only that budget exhaustion prevented refining all necessary facets, yet the returned boundary remains valid (envelope at any time) and includes the tightest gap leaf node that obstructs decision-making. Experience indicates that the current case studies focus on *compiler correctness and task feasibility* (local refinement unit accuracy, monotonic tightening curves). However, we acknowledge that the paper should report the frequency of UNKNOWN occurrences and their correlation with budget/specifications/networks.

---

### Author Response · Authors · 2025-12-03
**Summary for AC (rebuttal + reviewer discussion)**

**Overall paper & contribution (as understood by reviewers):**
 The submission introduces **JIT‑SWT**, a just‑in‑time semantics for ReLU/Max “CPWL” networks: compile the network into a guarded transducer with a *shared guard library* and a *shared expression DAG*, then **refine guards on demand** (rather than enumerating regions) while maintaining **sound anytime LB/UB envelopes**. Multiple downstream tasks (exact/certified Lipschitz, robustness/spec checking, equivariance/equivalence, causal influence) reduce to local LP/SOCP on refined cells.

------

## Review consensus & key points raised

**Strengths (broad agreement across reviewers):**

- The *core idea is principled* and targets the classic exponential blow‑up of eager region/materialization.
- The “shared carrier” concept is attractive: one representation supporting multiple geometry/verification/causal tasks.
- Theoretical properties (sound anytime envelopes; monotone progress; complexity tracking in #splits/#faces) were viewed positively (esp. WzSZ, 4K8x, m6BL).

**Primary weaknesses (repeated across reviews):**

1. **Presentation/readability is poor** (R9ZJr, m6BL, 4K8x): main text reads like a theorem catalog; insufficient intuition/roadmap/examples.
2. **Novelty delineation unclear** (R9ZJr): AF‑series looks like CPWL preliminaries; unclear what is foundational vs new.
3. **Empirical evaluation lacks baselines / limited scale** (R9ZJr, WzSZ, 4K8x, m6BL): no comparisons to verification tools or VNN‑COMP style benchmarks; experiments are small, so practical value/scalability remains uncertain.
4. **Decidability/UNKNOWN practicality** (R9ZJr, WzSZ): DYN‑6 depends on finite refinement + fair scheduling; reviewers asked how often UNKNOWN occurs under realistic budgets and sensitivity to scheduling policies.
5. **Scope limitation to CPWL** (WzSZ): modern architectures have non‑CPWL modules; how would the framework extend?

**Scores/leanings:**

- One strong reject (R9ZJr, confident 5) mainly due to presentation + missing baselines/literature.
- Three “borderline accept / would not mind reject” (WzSZ, 4K8x, m6BL), generally positive on theory/idea but concerned about clarity + evaluation + scalability evidence.

------

## Authors’ rebuttal: what they addressed

**(A) Roadmap + novelty separation:**

-  We explicitly state **AF‑1..5 are background CPWL facts**, while the **novel core starts at SWT‑1 and DYN‑1..6** (shared-guard SWT compilation + JIT refiners + anytime envelopes + dominance pruning + budgeted complexity + decidability statement). We claim to add a **contribution map (Fig.1) and theorem index (Table 1)** to connect AF→SWT→DYN→(GEO/VER/CAU).

**(B) Readability improvements (claimed):**

- “Major narrative pass,” moved proofs to appendix, added an end‑to‑end pseudocode overview (Alg.1), and more intuition before definitions.

**(C) Scalability concerns (response stance):**

- We **do not claim** ResNet/Transformer‑scale “exact decidability” in this paper.
- We emphasize cost is proportional to **visited leaves** and controlled by budgets; they mention caching and a matrix‑free conv Jacobian option.

**(D) Related work + baselines:**

- We acknowledge citations were insufficient and claim to expand post‑2020 citations + qualitative positioning.
- **Crucially, they explicitly decline to provide VNN‑COMP / tool baseline comparisons** (Marabou, αβ‑CROWN, etc.) due to time constraints, arguing the goal is not an end‑to‑end verifier but a reusable semantic carrier.

**(E) UNKNOWN frequency under budgets:**

- We reiterate tri‑valued design: UNKNOWN means budget exhausted but envelopes remain sound, and they return the “tightest‑gap” leaf.
- We **admit** the paper should report UNKNOWN frequency and its correlation with budgets/specs, but do not provide new quantitative evidence in the rebuttal.

**(F) Extending beyond CPWL / comparisons to incomplete bounds:**

- For non‑CPWL modules (softmax/attention/norm), they propose using **certified linear/SOC outer relaxations as leaf oracles**, preserving sound bounds but losing “final exactness.”
- We claim the oracle is “pluggable,” so one could insert αβ‑CROWN‑style bounds at leaves when only fast certification is desired.

------

## What remains unresolved / decision-relevant

1. **Missing empirical baselines remain the main open issue.** Authors explicitly refuse benchmark-style comparisons, which leaves practical competitiveness/impact ambiguous for a verification-focused venue.
2. **Scalability evidence remains limited.** We reposition the contribution as “semantic carrier + on-demand refinement,” but do not empirically validate beyond small MNIST/CIFAR-subset/Karate cases.
3. **UNKNOWN rate and scheduler sensitivity remain largely unquantified.** Authors acknowledge this gap but do not close it with data.
4. **Readability likely improved**, and novelty separation is now clearer (via map/index + moving proofs), but this is contingent o

---

### Meta-Review · Area_Chair_QvvQ · 2026-01-07

**Summary:**

This paper presents a just-in-time piecewise-linear semantics for compiling ReLU-type networks into a guarded CPWL transducer with shared guards. Different from many prior neural network verification approaches that target specific tasks/properties, the proposed approach supports multiple downstream tasks with shared structure and incremental refinement. Reviewers acknowledged the theoretical contributions of the work (e.g., on JIT compilation and support of multiple downstream tasks), but also raised major concerns on paper presentation, computational scalability, and lack of qualitative and empirical comparison with existing literature.

**Reviewer Concerns:**

The concerns from the reviewers center around the poor paper presentation, the lack of empirical studies and especially comparison with existing methods, and the limitations on scalability and practicality. While the authors addressed some of those concerns, e.g., with improvement on readability and addition of more literature, the main concerns on lacking empirical results (especially comparison with existing methods) and more studies on scalability/practicality remain.

**Reviewer Scores:**

The reviewers did not respond to the rebuttal. Given the remaining concerns, it is unlikely that they would increase their ratings.

---

### Decision · Program_Chairs · 2026-01-26

Reject